# FORTE : FINDING OUTLIERS WITH REPRESENTATION TYPICALITY ESTIMATION

**Debargha Ganguly**[*1]**, Warren Morningstar**[2]**, Andrew Yu**[1]**, Vipin Chaudhary**[1]
[1]Case Western Reserve University, Cleveland, OH, USA
[2]Google Research, Mountain View, CA, USA
`{debargha,asy51,vipin}@case.edu, wmorning@google.com`

## ABSTRACT

Generative models can now produce photorealistic synthetic data which is virtually indistinguishable from the real data used to train it. This is a significant evolution over previous models which could produce reasonable facsimiles of the training data, but ones which could be visually distinguished from the training data by human evaluation. Recent work on OOD detection has raised doubts that generative model likelihoods are optimal OOD detectors due to issues involving likelihood misestimation, entropy in the generative process, and typicality. We speculate that generative OOD detectors also failed because their models focused on the pixels rather than the semantic content of the data, leading to failures in near-OOD cases where the pixels may be similar but the information content is significantly different. We hypothesize that estimating typical sets using self-supervised learners leads to better OOD detectors. We introduce a novel approach that leverages representation learning, and informative summary statistics based on manifold estimation, to address all of the aforementioned issues. Our method outperforms other unsupervised approaches and achieves state-of-the art performance on well-established challenging benchmarks, and new synthetic data detection tasks. Our code is available at github.com/DebarghaG/forte.

## 1 INTRODUCTION

In the past decade, deep learning has made significant strides, primarily due to the availability of large-scale annotated datasets (Deng et al., 2009) used in supervised learning and the emergence of self-supervised learning utilizing vast web-scale crawled open data (Schuhmann et al., 2022; Gao et al., 2020; Sharma et al., 2018; Radford et al., 2021b). The transition from large-scale annotated datasets to self-supervised learning was driven by the expensive and labor-intensive nature of creating these datasets and yet the concerns surrounding data usage rights persist (He et al., 2022b; Carlini et al., 2021; Huang et al., 2022). Recent advancements have led to the development of generative models that excel in generating highly realistic and detailed synthetic images (Stein et al., 2024). In this paper, we investigate identifying out-of-distribution (OOD) data and generated synthetic data created using large-scale pre-trained generative models, commonly referred to as "foundation models"(Bommasani et al., 2021).

Broadly speaking, data encountered during deployment that was not sampled from the distribution used to generate training data is considered OOD. OOD data represents a challenge to safe deployment of predictive models because they can make confident incorrect predictions, leading to actions which could have negative outcomes. Foundation models complicate the traditional definition of OOD slightly. These models are trained on extensive and diverse datasets, making the data generating process, and thus possible OOD inputs difficult to specify cleanly. Despite its diversity, the data generating process only samples a small portion of the input space, leaving many potential subspaces open for OOD contamination. This contamination can erode the calibration of predictive models trained using the foundation model as a base, representing a significant hazard for safe model deployment.

---

[*]This research was supported in part by NSF Awards 2117439 and 2112606.

Predictive model failures due to OOD inputs can be understood through the lens of typicality. The concept of typicality arises from information theory and codifies the difference between likelihood from a generative process, and the probability of generating a sample from that process with a particular likelihood. In low dimensions these two are generally equivalent, but this is often not the case in high dimensions (Nalisnick et al., 2019; Choi et al., 2018). The asymptotic equipartition property asserts that most of the probability mass of a distribution is contained in the region of high typicality, referred to as the "typical set" of a distribution (Cover, 1999). Thus, OOD inputs confound predictive models because the model has no incentive to make calibrated predictions on this data, since any errors in the predictive model for OOD inputs do not contribute to the risk to the hypothesized model, which is being minimized via training. At the same time, the atypicality of OOD data presents a potential way to identify them (Nalisnick et al., 2020; Morningstar et al., 2021), and thus avoid making risky predictions.

Measuring typicality directly is challenging because one cannot typically query the probability of an observed datum under the (unknown) data generating process. This has led many prior works to propose using generative models to approximate the process and use it to measure the probability, and therefore test the typicality of the input. However, Zhang et al. (2021) and Caterini & Loaiza-Ganem (2022) have pointed out challenges in this approach that arise due to likelihood misestimation, which can cause OOD inputs to have likelihoods under the generative model that are consistent with ID data.

In this paper, we hypothesize that many of the shortcomings with typicality-based approaches could be addressed using statistics which tune to the semantic content of the data. We propose to leverage self-supervised representations, which extract semantic information while discarding many potential confounding features (e.g. textures, backgrounds). Our specific contributions are:

1. Forte, a novel framework combining diverse representation learning techniques (CLIP, ViT-MSN, and DINOv2) for typicality estimation, uses both parametric (GMM) and non-parametric (KDE, OCSVM) density-based methods to detect atypical samples, i.e., out-of-distribution (OOD) and synthetic images generated by foundation models. Forte requires no class labels, exposure to OOD data during training, or restrictions on the architecture of generative models.

2. A set of per-point summary statistics (precision, recall, density, and coverage) that effectively capture the "probability distribution of the representations" using reference and unseen test samples in the feature space, enabling more nuanced anomaly detection.

3. Extensive experiments demonstrating Forte's superior performance compared to state-of-the-art supervised and unsupervised baselines on various OOD detection tasks, and synthetic image detection, including photorealistic images generated by advanced techniques like Stable Diffusion.

4. Insights into the limitations of relying solely on statistical tests and distribution-level metrics for assessing the similarity between real and synthetic data, highlighting the effectiveness of our proposed per-point summary statistics and anomaly detection framework.

## 2 RELATED WORKS

In discriminative machine learning, the assumption that inference data mirrors the training data distribution is foundational, yet often flawed. The occurrence of out-of-distribution (OOD) inputs can lead to misleadingly confident but incorrect predictions by models, posing significant reliability and safety concerns. Large neural networks are vulnerable to adversarial perturbations (Szegedy et al., 2013) and poor calibration (Guo et al., 2017), necessitating OOD detection. OOD detection methods are either supervised, using labels and examples to calibrate models or train them to identify OOD data (Liang et al., 2018; Hendrycks et al., 2019b; Meinke & Hein, 2020; Dhamija et al., 2018), or unsupervised, employing generative models to approximate the training data density $q(\mathbf{X})$ and determine prediction reliability via $q(Y|\mathbf{X})$, assuming OOD inputs have lower probability (Bishop, 1994).

**Unsupervised OOD detection methods** typically use generative models to measure the likelihood of the data. (Bishop, 1994) assumed low likelihood would be observed on OOD data, but this may fail in high dimensionality (Choi et al., 2018; Nalisnick et al., 2019; Hendrycks et al., 2019b; Serrà

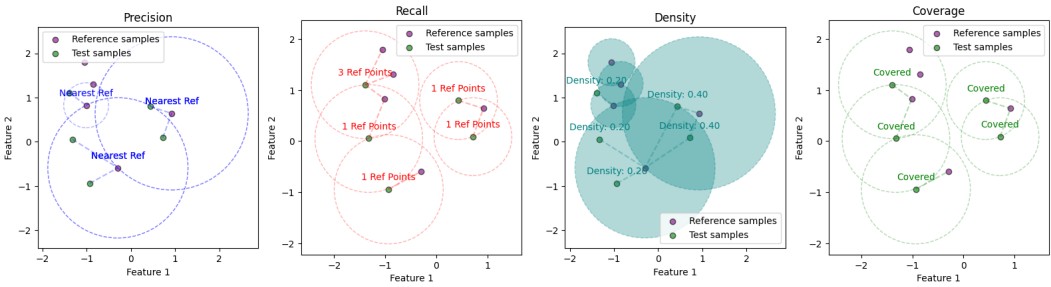

Figure 1: Visualization of Precision, Recall, Density, and Coverage metrics for reference and test samples in a 2D feature space, with nearest neighbour k=1. Precision is depicted by blue circles around reference samples, recall by red circles around test samples, density by solid teal circles around reference samples, and coverage by green circles around test samples.

et al., 2019). Fixes include using WAIC (Choi et al., 2018), likelihood ratios (Ren et al., 2019), or typicality tests (Nalisnick et al., 2019), but these have limitations associated with high dimensional likelihoods. Morningstar et al. (2021) proposed the Density of States Estimator (DoSE). Inspired by principles from statistical physics, it leverages multiple summary statistics from generative models to distinguish between in-distribution and OOD data.

**Supervised OOD detection** leverages labeled in-distribution and/or known OOD data. Techniques include using VIB rate (Alemi et al., 2018), calibrating predictions with ODIN (Liang et al., 2018) or its generalization (Hsu et al., 2020), ensembling classifiers (Lakshminarayanan et al., 2017), training with OOD examples to encourage uniform confidence (Hendrycks et al., 2019b), adversarial attacks to lower confidence near training data (Stutz et al., 2019), visual concept networks to model human understandable features in graphs (Ganguly et al., 2025) and fitting Gaussian mixtures to latent representations (Meinke & Hein, 2020). While effective, these methods require labels or specific outliers.

**Generative model assessment** can be done over four categories as shown by Stein et al. (2024): *ranking*, *fidelity*, *diversity*, and *memorization*. Ranking metrics include FD (Heusel et al., 2017), which measures the Wasserstein-2 distance between real and generated distributions; $FD_\infty$ (Chong & Forsyth, 2020), which removes bias due to finite sample size; sFID (Nash et al., 2021), which uses a spatial representation; KD (Bińkowski et al., 2018), a proper distance between distributions; IS (Salimans et al., 2016), which evaluates label entropy and distribution; and FLS (Jiralerspong et al., 2023), a density estimation method. Fidelity and diversity are measured using precision, recall (Sajjadi et al., 2018; Kynkäänniemi et al., 2019b), density, coverage (Naeem et al., 2020a), rarity score (Han et al., 2022), and Vendi score (Friedman & Dieng, 2022), which quantify the quality and variety of generated samples. Memorization is quantified using AuthPct (Alaa et al., 2022), $C_T$ score (Meehan et al., 2020), FLS-POG (Jiralerspong et al., 2023), and memorization ratio with calibrated $l_2$ distance (Somepalli et al., 2022), which detect overfitting and copying of training data. These metrics can be computed in various supervised (including Inception-V3 (Szegedy et al., 2016)) and self-supervised representation spaces, including self-supervised model families like contrastive (SimCLRv2 Chen et al. (2020)), self-distillation (DINOv2 Oquab et al. (2023a)), canonical correlation analysis (SwAV Caron et al. (2020)), masked image modelling (MAE He et al. (2022a), data2vec Baevski et al. (2022)), DreamSim Fu et al. (2023), and language-image (CLIP Radford et al. (2021a), sometimes using the OpenCLIP implementation Ilharco et al. (2021) trained on DataComp-1B Gadre et al. (2023)).

# 3 FORTE: A FRAMEWORK FOR OOD DETECTION USING PER-POINT METRICS

In this section, we introduce Forte, a novel framework that combines diverse representation learning techniques with per-point summary statistics and non-parametric density estimation models to detect out-of-distribution (OOD) and synthetic data.

## 3.1 PROBLEM SETUP

We start with a dataset $X = \{x_i^r\}_{i=1}^m$ that comes from a true but unknown distribution $p$, where each $x_i \in \mathbb{R}^d$ is sampled independently and identically from $p$. In the context of out-of-distribution detection, we consider that the unseen data $\{x_j^g\}_{j=1}^n$ might come from a mix of the true distribution $p$ and an unknown confounding distribution $\tilde{p}$ (which can be from an OOD benchmark or synthetic data generated using a model such as Stable Diffusion). The mixed distribution is denoted by $\dot{X} \sim \alpha p(\dot{X}) + (1 - \alpha)\tilde{p}(\dot{X})$, where $\alpha$ is an unknown mixing parameter. Since both $\alpha$ and $\tilde{p}$ are unknown, we cannot directly obtain samples from $\tilde{p}$ or make any assumptions about $\alpha$ and $\tilde{p}$. The goal of OOD detection is to develop a decision rule to determine when the input data $\dot{X}$ is atypical.

We build on Density of States Estimation (DoSE; Morningstar et al., 2021), which is the SoTA unsupervised method that addresses the OOD detection problem. The first step in DoSE is to create a *summary statistic* $T_n$ that can help evaluate new, unseen data. Some examples of $T_n$ are the negative log-likelihood of $X$, the $L_2$ distance between $X$ and the sample mean, or the maximum likelihood estimate of a joint distribution $q(Y|X, \theta_n)$. To determine when the input data $\dot{X}$ is atypical, DoSE models the distribution of $T$ values in the training data and uses the density to score the typicality of inputs based on the measured values of their statistics. While DoSE gives a valuable mechanism by which one can determine the typicality of a particular input, it does not identify which statistics are informative. DoSE further limited its focus to likelihood-based generative models, which Caterini & Loaiza-Ganem (2022) showed may be suboptimal for OOD detection.

## 3.2 CREATING GENERALIZED NOVEL SUMMARY STATISTICS

We propose the following *per-point metrics*, novelly adopted to OOD detection from previous work in manifold estimation inside representation spaces by Naeem et al. (2020b) and Kynkäänniemi et al. (2019a) for capturing different facets of the generated samples. Let $\mathbb{1}(\cdot)$ be the indicator function, $S(\{x_i^r\}_{i=1}^m) = \bigcup_{i=1}^m B(x_i^r, \mathrm{NND}_k(x_i^r))$, where $B(x, r)$ is a Euclidean ball centered at $x$ with radius $r$, and $\mathrm{NND}_k(x_i^r)$ is the distance between $x_i^r$ and its $k$-th nearest neighbor in $\{x_i^r\}_{i=1}^m$, excluding itself.

1. **Precision per point** is a binary statistic that indicates whether each test point is within the nearest neighbor distance of any reference point:

$$\mathrm{precision}_{\mathrm{pp}}^{(j)} = \mathbb{1}\left(x_j^g \in S(\{x_i^r\}_{i=1}^m)\right). \tag{1}$$

   A high overall precision indicates that the test samples are closely aligned, and similar to the reference data distribution.

2. **Recall per point** is computed for each test point by counting the number of reference points within its nearest neighbor distance and dividing by the total number of reference points:

$$\mathrm{recall}_{\mathrm{pp}}^{(j)} = \frac{1}{m} \sum_{i=1}^m \mathbb{1}\left(x_i^r \in B(x_j^g, \mathrm{NND}_k(x_j^g))\right). \tag{2}$$

   A high recall implies that the test distribution collectively covers a significant portion of the reference data distribution, i.e. that the test distribution contains diverse samples that represent the different regions of the reference data distribution.

3. **Density per point** is computed for each test point by counting the number of reference points for which it is within that points nearest neighbor distance and dividing by the product of $k$ and the total number of reference points:

$$\mathrm{density}_{\mathrm{pp}}^{(j)} = \frac{1}{km} \sum_{i=1}^m \mathbb{1}\left(x_j^g \in B(x_i^r, \mathrm{NND}_k(x_i^r))\right). \tag{3}$$

   A test point with high density suggests that it is located in a region of high probability density in the reference data distribution, i.e. it captures the underlying density of the reference data. Density per point essentially measures the expected likelihood of test points against the reference manifold, providing a more informative measure than the binary precision per point.

4. **Coverage per point** is computed for each test point by checking if its distance to the nearest reference point is less than its own nearest neighbor distance:

$$\text{coverage}_{\text{pp}}^{(\text{j})} = \mathbb{1}\left(\min_i(d(x_j^g, x_i^r)) < \text{NND}_k(x_j^g)\right). \tag{4}$$

High coverage indicates that the test samples are well-distributed and cover the support of the reference data distribution. Coverage per point effectively improves upon the original recall metric by building manifolds around reference points, making it more robust to outliers in the test data and computationally efficient.

These per-point metrics serve as summary statistics that capture local geometric properties of the data manifold in the feature space. They enable us to model the distribution of in-distribution (ID) data and identify out-of-distribution (OOD) samples effectively. We extract representations from reference and test images using CLIP Radford et al. (2021b), which learns to map images and text into a shared latent space, self-supervised models Gui et al. (2023) like ViT-MSN Assran et al. (2022), which predicts similarity between masked views of the same image, and DiNo v2 Oquab et al. (2023a), which distills knowledge from a teacher to a student network. Combining representations from these diverse models captures distinct aspects of the data via different manifolds and improves robustness of our anomaly detection approach, as we observe later in Table 3.

## 3.3 DEVELOP DECISION RULE

We use the summary statistics to develop an non-parametric density estimator as an anomaly detection model. First, we generate feature vectors for the reference data using self-supervised learning methods. We split the data into three parts: one-third for held-out testing, one-third as the reference distribution, and one-third as a test distribution that is drawn from the reference distribution. We calculate summary statistics for the second and third to understand what these statistics look like when the test data matches the real data (i.e. $P \overset{d}{=} Q$). We train One-Class SVM (Schölkopf et al., 2001), Gaussian Kernel Density Estimation (Parzen, 1962), and Gaussian Mixture Model (Reynolds et al., 2009) on the reference summary statistics. These models learn a decision boundary enclosing the typical set of the reference data distribution. We then test the models' ability to distinguish between a mix of held-out test and reference features by comparing their atypicality i.e. summary statistics to the reference distribution. By evaluating the models' performance on the test set, we assess their effectiveness in detecting OOD samples and distinguishing real from synthetic data distributions.

**To assess the performance** of the anomaly detection models, using an unseen part of the training real distribution and the generated data distribution, we use Area Under the Receiver Operating Characteristic Curve (AUROC), which measures the ability of the model to discriminate between normal and anomalous points across different decision thresholds, and False Positive Rate at 95% True Positive Rate (FPR@95) which indicates the proportion of normal points incorrectly identified as anomalies when the model correctly identifies 95% of the true anomalies.

Under certain theoretical assumptions, we can justify the effectiveness of our per-point metrics in distinguishing between in-distribution (ID) and out-of-distribution (OOD) data. Specifically, if the reference data $\{x_j^r\}_{j=1}^m$ and the test data $\{x_i^g\}_{i=1}^n$ are drawn from Gaussian distributions with the same covariance but different means (with a significant mean difference), the expected values of these metrics differ markedly between ID and OOD samples. For ID data, the expected per-point precision and coverage approximate $1 - e^{-k}$, the expected per-point recall is roughly $k/m$, and the expected per-point density is close to 1. In contrast, for OOD data, these expected values are near zero due to the large mean difference, which causes the test samples to fall outside the typical regions of the reference distribution. This substantial disparity provides a strong theoretical foundation for using these metrics as effective summary statistics for OOD detection. By computing these metrics for test samples and comparing them to the expected ID values, we can effectively identify anomalous data points. A detailed proof and further explanations are provided in the Appendix C.

## 4 EXPERIMENTS

**Unsupervised Baselines :** We train an ensemble of deep generative models on in-distribution data, validate on a heldout set to ensure no memorization, and compute DoSE scores on in-distribution

and OOD test sets. We measure OOD identification performance and compare against several unsupervised baselines: single-sided threshold (Bishop, 1994), single-sample typicality test (TT) (Nalisnick et al., 2019), Watanabe-Akaike Information Criterion (WAIC) (Choi et al., 2018), and likelihood ratio method (LLR) (Ren et al., 2019). Given the best demonstrated unsupervised OOD detection results were in (Morningstar et al., 2021), we stick to using Glow models (Kingma & Dhariwal, 2018), where we use summary statistics: log-likelihood, log-probability of the latent variable, and log-determinant of the Jacobian between input and transformed spaces.

**Supervised Baselines:** We also compare our approach against state-of-the-art methods in out-of-distribution (OOD) detection literature, selecting the *best reported result*s on established benchmarks, *regardless of the model architecture or techniques* employed. Specifically, we consider results from NNGuide (Park et al., 2023), Virtual Logit Matching (Wang et al., 2022), and OpenOOD v1.5 (Zhang et al., 2024a). For OpenOOD v1.5, we select the top-performing entries from the leaderboard, which utilize a Vision Transformer (ViT-B) trained with cross-entropy loss, along with Maximum Logit Score (MLS) (Hendrycks et al., 2019a) and Relative Mahalanobis Distance Score (RMDS) (Ren et al., 2021) for OOD detection. To explore the potential of foundation models in OOD detection, OpenOOD also employs a linear probing of Dinov2 (Oquab et al., 2023b) in conjunction with MLS, however performance is demonstrated to be poor.

## 4.1 Baseline Performance for Synthetic Data Distributions

**Distribution Divergence:** We compute pairwise distances and divergences between real and generated feature distributions. First we start with KL Divergence which measures information loss when approximating $P$ with $Q$ Kullback & Leibler (1951): $D_{\text{KL}}(P \parallel Q) = \sum_{x \in \mathcal{X}} P(x) \log \frac{P(x)}{Q(x)}$, and JS Divergence which is the symmetric KL divergence variant, comparing distributions with disjoint support Lin (1991): $D_{\text{JS}}(P \parallel Q) = \frac{1}{2} \left( D_{\text{KL}}(P \parallel M) + D_{\text{KL}}(Q \parallel M) \right), M = \frac{1}{2}(P + Q)$. The Wasserstein Distance on the other hand gives the minimum cost to transform one distribution into the other Rubner et al. (2000): $W(P, Q) = \inf_{\gamma \in \Gamma(P,Q)} \mathbb{E}(x, y) \sim \gamma[\|x - y\|], \Gamma(P, Q)$: joint distributions with marginals $P, Q$, while the Bhattacharyya Distance gives the similarity between $P$ and $Q$, 0 (identical) to $\infty$ (separated) Bhattacharyya (1943): $D_B(P, Q) = -\ln \left( \sum x \in \mathcal{X} \sqrt{P(x)Q(x)} \right)$

**CLIP-based Zero Shot Strong OOD Baseline:** To establish baseline performance for detecting anomalous generated images directly from representations, we split the in-distribution reference features $X^r$ extracted from any of the feature extraction model (e.g. CLIP) into training $X^r_{\text{train}}$ and testing $X^r_{\text{test}}$ sets and train all the same non-parametric density estimation models with the same hyperparameter tuning, including OCCSVM, KDE and GMM. We evaluate each model on a test set consisting of representations from held-out reference images $X^r_{\text{test}}$ and anomalous images $X^g$. This provides an strong initial assessment of how well these models can distinguish between reference and anomalous data based on, for e.g. the CLIP features.

**Statistical Tests:** To rigorously compare the real and generated feature distributions, we perform several statistical tests. The first is the Two-sample Kolmogorov-Smirnov (KS) Test, which is a non-parametric test that compares the empirical Cumulative Distribution Functions (CDFs) of two samples. The KS statistic measures the maximum distance between the CDFs, with a corresponding p-value for the null hypothesis that the samples are drawn from the same distributionMassey Jr (1951). The Mann-Whitney U Test gives a non-parametric test for whether two independent samples are drawn from the same distribution. It is based on the ranks of the observations in the combined sampleMann & Whitney (1947). The Z-test compares the standardized differences between the means of two samples, assuming they are normally distributed. The Z-score measures the distance between the sample means in units of standard error Sprinthall (2011). Finally we also calculate the Anomaly Detection with Local Outlier Factor (LOF) and Isolation Forest (IF): These algorithms assign anomaly scores to each point based on their local density (LOF) Breunig et al. (2000) or ease of isolation (IF) Liu et al. (2018) relative to the real data. Higher scores indicate a generated image is anomalous.

**Generative Evaluation Techniques:** In line with suggestions from (Stein et al., 2024), we use the Fréchet distance (FD) and $FD_\infty$ computed with the DINOv2 encoder, instead of the inception network to tell how different the generated images are from the reference set of images. We also use, the CLIP maximum mean discrepancy (CMMD), which is a distance measure between probability

Table 1: Performance comparison between our method and established supervised SoTA methods on various out-of-distribution (OOD) detection tasks. Our method consistently achieves the best performance across a range of dataset pairings, particularly outperforming on challenging datasets like NINCO and SSB-Hard. It also excels at detecting covariate shifted ID datasets. Key configurations include ViT-B trained with Cross Entropy and the RMDS postprocessor, and DINOv2+MLS, which utilizes training with Linear Probe DINOv2 alongside the MLS postprocessor. For reliable measurements, Forte is run with 10 random seeds. More comparisons are provided in Fig. 16

| | | NNGuide | | ViM | | ViT-B+CE+RMDS | | DINOv2+MLS | | Forte+GMM(Ours) | |
|---|---|---|---|---|---|---|---|---|---|---|---|
| **In-Distr (ImageNet-1k)** | | AUROC | FPR95 | AUROC | FPR95 | AUROC | FPR95 | AUROC | FPR95 | AUROC | FPR95 |
| | iNaturalist | 99.57 | 1.83 | 99.41 | 2.60 | 96.10 | 19.47 | 98.41 | 5.64 | $99.67 \pm 00.03$ | $0.64 \pm 00.06$ |
| | Texture | 95.82 | 21.58 | 95.34 | 20.31 | 89.38 | 37.22 | 91.82 | 33.95 | $98.04 \pm 00.10$ | $5.61 \pm 00.25$ |
| **Far-OOD** | OpenImage-O | | | | | 92.32 | 29.57 | 95.58 | 16.88 | $96.73 \pm 00.11$ | $11.77 \pm 00.57$ |
| | NINCO | | | | | 87.31 | 46.20 | 88.38 | 41.02 | $98.34 \pm 00.09$ | $5.18 \pm 00.51$ |
| **Near-OOD** | SSB-Hard | | | | | 72.87 | 84.52 | 77.28 | 72.90 | $94.95 \pm 00.17$ | $22.30 \pm 01.45$ |
| | ImageNet-C | | | | | | | | | $82.25 \pm 00.16$ | $42.56 \pm 02.23$ |
| | ImageNet-R | | | | | | | | | $93.60 \pm 00.66$ | $20.68 \pm 01.83$ |
| **Covariate ID** | ImageNet-V2 | | | | | | | | | $59.28 \pm 00.21$ | $90.40 \pm 00.34$ |

Table 2: Comparison of performance figures between our method and various unsupervised baselines for out-of-distribution (OOD) detection tasks, demonstrating superior performance of our method across all challenging dataset pairings. For reliable measurements, Forte is run with 10 random seeds. This is presented as a visualization in Fig.15

| Method | In-Dist (CIFAR-10) | OOD (CIFAR-100) | OOD (Celeb-A) | OOD (SVHN) |
|---|---|---|---|---|
| $q(X|\theta_n)$ | AUROC | 52.00 | 91.40 | 6.50 |
| | FPR95 | 91.67 | 51.10 | 100.0 |
| **WAIC** | AUROC | 53.20 | 92.80 | 14.30 |
| | FPR95 | 90.16 | 46.04 | 98.83 |
| **TT** | AUROC | 54.80 | 84.80 | 87.00 |
| | FPR95 | 92.06 | 74.00 | 61.28 |
| **LLR** | AUROC | 48.13 | 80.08 | 64.21 |
| | FPR95 | 93.44 | 64.79 | 76.36 |
| **DoSE** | AUROC | 56.90 | 97.60 | 97.30 |
| | FPR95 | 91.95 | 12.82 | 13.16 |
| **Forte+SVM (Ours)** | AUROC | $97.29 \pm 00.55$ | $100.00 \pm 00.00$ | $99.84 \pm 00.05$ |
| | FPR95 | $09.08 \pm 01.82$ | $0.00 \pm 00.00$ | $00.00 \pm 00.00$ |
| **Forte+KDE (Ours)** | AUROC | $94.81 \pm 01.22$ | $99.75 \pm 00.06$ | $98.37 \pm 02.07$ |
| | FPR95 | $18.20 \pm 03.82$ | $0.06 \pm 00.07$ | $07.53 \pm 10.77$ |
| **Forte+GMM (Ours)** | AUROC | $97.63 \pm 00.15$ | $100.00 \pm 00.00$ | $99.49 \pm 00.48$ |
| | FPR95 | $09.69 \pm 01.08$ | $0.00 \pm 00.00$ | $0.00 \pm 00.00$ |

distributions that presents benefits over the Fréchet distance. With an appropriate kernel, CMMD does not assume any specific distribution, unlike the Fréchet distance which assumes multivariate normal distributions.

## 5 RESULTS

Tables 1 & 2 present the performance comparison between our proposed methods (Forte+SVM, Forte+KDE, and Forte+GMM) and various state of the art supervised and unsupervised techniques for out-of-distribution (OOD) detection techniques. Our methods consistently outperform techniques across all challenging dataset pairings, demonstrating their effectiveness in detecting OOD samples without relying on labeled data.

Table 3 presents an ablation study investigating the impact of incorporating multiple representation learning techniques into the Forte framework, when detecting arbitrary superclasses from the ImageNet1k hierarchy. In this challenging task example, the in-distribution data consists of various vehicle classes from the ImageNet dataset, such as Ambulance, Beach wagon, Cab, Convertible,

Table 3: Ablation study investigating the impact of incorporating multiple representation learning techniques for detecting arbitrary superclasses from the ImageNet1k hierarchy, using Forte+GMM.

| In Distribution (Superclass ImageNet 2012) | Far OOD | | Near OOD | |
|---|---|---|---|---|
| | AUROC | FPR95 | AUROC | FPR95 |
| Forte (CLIP) | 99.42 | 00.04 | 88.84 | 61.04 |
| Forte (MSN) | 99.13 | 00.39 | 82.63 | 77.53 |
| Forte (DINOv2) | 99.79 | 00.03 | 86.97 | 79.25 |
| Forte (CLIP+MSN) | 99.96 | 00.02 | 90.00 | 30.77 |
| Forte (CLIP+DINOv2) | 99.98 | 00.01 | **91.35** | **26.89** |
| Forte (DINOv2+MSN) | 99.97 | 00.00 | 88.71 | 31.65 |
| Forte (all 3) | **100.00** | **00.01** | 91.17 | 28.91 |

Jeep, Limousine, Minivan, Model t, Racer, and Sportscar. The out-of-distribution data is divided into two categories: Near OOD, which includes other utility vehicle classes like fire engine, garbage truck, pickup, tow truck, trailer truck, minivan, moving van, and police van; and Far OOD, which consists of animal classes such as Egyptian cat, Persian cat, Siamese cat, Tabby cat, Tiger cat, Cougar, Lynx, Cheetah, Jaguar, Leopard, Lion, Snow leopard, and Tiger. Our method is successful in very challenging near-OOD situations, and becomes more so when using multiple representations. The combination of all three techniques (CLIP, MSN, and DINOv2) achieves the best overall performance for both Far OOD detection, and comes a close second in Near OOD detection. These findings suggest that leveraging diverse SSL representation techniques within the Forte framework can capture complementary information and enhance the overall OOD detection capabilities.

## 5.1 PERFORMANCE ON SYNTHETIC DATA DETECTION

**To generate synthetic data** for assessing distributional robustness, we first employ the Stable Diffusion Img2Img setting (Rombach et al., 2022), where a diffusion-based model can generate new images conditioned on an input image and a text prompt. We use the Stable Diffusion 2.0 base model and generate images with varying strength parameters (0.3, 0.5, 0.7, 0.9, 1.0) to control the influence of the input image on the generated output, essentially allowing our real distribution to be prior of controllable strength for the generated distribution. We also use the Stable Diffusion 2.0 text-to-image model to generate new images conditioned on the captions generated by BLIP(Li et al., 2022) for each real image from the reference distribution. This allows us to create images that are semantically similar to the real images but with novel compositions and variations. Finally, we also generate images directly from the class name associated with each real image (e.g., "a photo of a {monarch butterfly}, in a natural setting"). This provides a baseline for generating images that capture the essential characteristics of the class without relying on specific input images or captions. The image generation pipeline is implemented using the Hugging Face Transformers (Wolf et al., 2020) and Diffusers (von Platen et al., 2022) libraries, which provide high-level APIs for working with pre-trained models. Examples can be found in Figure 2, Figure 3 & Figure 4.

**Fréchet Distance (FD), $FD_\infty$, and CMMD scores** provide insights into the distribution shift between real and synthetic images, with generated images moving further away from the real distribution as diffusion strength increases. However, these distribution-level statistics do not provide information about individual images within the distribution, necessitating an OOD detection strategy. Tables 4 & 7 compare the performance of Forte+GMM against a strong CLIP-based baseline on the Golden Retriever and Volleyball classes from ImageNet. For the well-represented Golden Retriever class, Forte+GMM consistently outperforms the baseline across all image generation settings, achieving near-perfect AUROC scores and low FPR95 values for Img2Img with strength parameters 0.9 and 1.0, Caption-based, and Class-based image generation. Lower performance at lower strengths is due to images being too similar to the real distribution. The Volleyball class, part of Hard ImageNet (Moayeri et al., 2022), focuses on classes with strong spurious cues. Volleyballs rarely occur alone in ImageNet images, and generating images without appropriate priors results in mode collapse (see Figure 4). Forte+GMM surpasses the CLIP-based baseline in all image generation scenarios, with notable improvements for Img2Img with higher strength parameters, Caption-based, and Class-based image generation (AUROC scores > 97%, FPR95 values < 6%).

Table 4: Performance comparison of Forte+GMM against a CLIP-based baseline, and Gen Eval methods for detecting synthetic images of Golden Retrievers generated using various techniques.

| | FD | $FD_\infty$ | CMMD Score | Baseline Model CLIP | | Forte+GMM | |
|---|---|---|---|---|---|---|---|
| | | | | AUROC | FPR95 | AUROC | FPR95 |
| **Img2Img S=0.3** | 453.63 | 418.22 | 0.52 | 61.19 | 86.27 | **68.28 $\pm$ 02.14** | **68.07 $\pm$ 05.56** |
| **Img2Img S=0.5** | 648.97 | 624.01 | 0.64 | 59.49 | 86.27 | **82.93 $\pm$ 02.50** | **46.80 $\pm$ 10.68** |
| **Img2Img S=0.7** | 762.17 | 735.42 | 0.64 | 61.23 | 86.36 | **94.19 $\pm$ 01.85** | **24.90 $\pm$ 12.99** |
| **Img2Img S=0.9** | 845.96 | 819.06 | 0.66 | 59.05 | 88.87 | **97.59 $\pm$ 01.23** | **14.41 $\pm$ 13.55** |
| **Img2Img S=1.0** | 891.39 | 870.17 | 0.73 | 60.15 | 89.23 | **98.11 $\pm$ 00.75** | **06.09 $\pm$ 05.72** |
| **Caption-based** | 575.18 | 546.42 | 0.90 | 80.71 | 71.54 | **96.77 $\pm$ 01.14** | **18.90 $\pm$ 14.38** |
| **Class-based** | 1,065.96 | 1,048.18 | 1.07 | 75.73 | 82.23 | **98.26 $\pm$ 01.12** | **10.22 $\pm$ 13.04** |

Tables 6 & 7 compare various statistical measures for assessing the similarity between distributions of real and generated images across different image generation methods. The results are shown for the Golden Retriever and Volleyball classes. For both classes, the statistical tests fail to provide clear and consistent indications of distributional differences between real and generated images. Z-scores are close to zero, suggesting no significant difference in means, while K-S test and Mann-Whitney U test p-values are inconsistent across generation methods. LOF and IF scores show limited variation and struggle to identify anomalies in the generated images. JSD, KLD, Wasserstein distance, and Mahalanobis distance do not exhibit clear trends as generation strength increases, making it difficult to draw meaningful conclusions about distributional similarity. To add to this problem, JSD and KLD are not defined for any distribution of representations apart from MSN embeddings. Bhattacharya distances are not defined on any representation, indicating no overlap. These results highlight the limitations of relying solely on statistical tests comparing representations of new points to the original distribution, as they often fail to capture subtle distributional differences between real and generated images, particularly when generated images are highly realistic. The inconsistencies and lack of clear trends underscore the need for more sophisticated approaches, such as the proposed Forte+GMM framework, which leverages diverse feature extraction techniques and anomaly detection algorithms to effectively detect synthetic images.

## 5.2 PERFORMANCE ON MEDICAL IMAGE DATASETS

Magnetic resonance imaging (MRI) datasets and models present a unique challenge in a high stakes scenario, making robust out-of-distribution (OOD) detection paramount. These datasets, typically acquired under specific study protocols, suffer from batch effects that hinder model generalization even when changes in protocols are minute (Horng, 2023). The problem is exacerbated by dataset homogeneity within studies and limited dataset sizes in clinical applications, making it impractical to train separate models for each batch. Subtle distribution shifts between datasets of the same subject matter, though not immediately apparent to human observers, can significantly impact model performance. While data augmentation and harmonization methods have been explored (Hu et al., 2023), existing metrics for detecting distribution drift and assessing harmonization effectiveness are limited. Our work proposes deploying Forte as a more robust metric to address this gap by effectively differentiating between datasets that may appear similar but have crucial distributional differences. Using public datasets like FastMRI Zbontar et al. (2018); Knoll et al. (2020) and the Osteoarthritis Initiative (OAI) Nevitt et al. (2006), we simulate realistic scenarios where models trained on one dataset (treated as in-distribution) are confronted with another (considered OOD). Table 5 demonstrates the effectiveness of Forte in differentiating the datasets. In other tests over closed-source but real-world hospital data, we observe similar near perfect performance, suggesting that Forte can have zero-shot applications in a variety of other sensitive domains.

## 6 DISCUSSION

While just using precision and density as summary statistics would lead Forte to be considered single-sample tests, recall and coverage make our framework a two-sample test, as they take into account

Table 5: Out-of-distribution (OOD) detection using Forte, applied to medical image datasets. Strong performance by Forte suggests the presence of batch effects and a need for data harmonization. Refer to Appendix E for more details on the FastMRI and OAI datasets. For reliable estimation, performance is measured over 10 random seeds.

| Method | Metric | In-Dist: OOD: | FastMRI NoFS OAI TSE | FastMRI FS OAI T1 | FastMRI FS OAI MPR |
|---|---|---|---|---|---|
| Forte+SVM (Ours) | AUROC | | $100.00 \pm 0.00$ | $100.00 \pm 0.00$ | $100.00 \pm 0.00$ |
| | FPR95 | | $00.00 \pm 00.00$ | $00.00 \pm 00.00$ | $00.00 \pm 00.00$ |
| Forte+KDE (Ours) | AUROC | | $97.97 \pm 5.63$ | $95.98 \pm 7.84$ | $95.99 \pm 7.86$ |
| | FPR95 | | $9.49 \pm 26.76$ | $19.75 \pm 39.10$ | $19.77 \pm 39.31$ |
| Forte+GMM (Ours) | AUROC | | $99.95 \pm 0.13$ | $99.91 \pm 0.16$ | $99.91 \pm 0.17$ |
| | FPR95 | | $0.00 \pm 0.00$ | $0.00 \pm 0.00$ | $0.00 \pm 0.00$ |

the relationship between each synthetic image and the entire set of real images. This also allows us to better understand the behavior of the generated images in relation to the real data distribution. This is usually not a problem in production, and benchmarks, since when you check whether a sample is out of distribution, you usually sample from the mixture probability distribution of in and out of distribution, which is what we really use here. Prior work has also operated in this paradigm, i.e. two sample OOD test such as the typicality test (Nalisnick et al., 2019).

When selecting non-parametric density estimators to model typicality, it is important to consider the manifold properties. We observe that GMM excels with clustered data, especially with a bigger number of gaussians when operating with data from multiple possible classes, while KDE struggles with sharp density variations. OCSVM is robust to outliers and performs well in high-dimensional spaces, making it suitable for cohesive normal data.

DoSE (Morningstar et al., 2021) pioneered chaining multiple summary statistics for typicality measurement, using ID sample distributions to construct typicality estimators rather than direct statistic values. While groundbreaking, DoSE's reliance on generative model likelihoods proved problematic, as subsequent work (Caterini & Loaiza-Ganem, 2022; Zhang et al., 2021) showed these can be unreliable for OOD detection. Our approach addresses these limitations through four key improvements: (1) utilizing self-supervised representations to capture semantic features, (2) incorporating manifold estimation to account for local topology, (3) unifying typicality scoring and downstream prediction models to minimize deployment overhead, and (4) eliminating additional model training requirements. These advances yield substantial empirical gains. While building upon DoSE's fundamental statistical machinery, our modifications dramatically enhance practical performance.

## 7 CONCLUSION

Our work underscores the importance of developing robust methods for detecting out-of-distribution (OOD) data and synthetic images generated by foundation models. The proposed Forte framework combines diverse representation learning techniques, non-parametric density estimators, and novel per-point summary statistics, demonstrating far superior performance compared to state-of-the-art baselines across various OOD detection tasks and synthetic image detection tasks. The experimental results not only showcase the effectiveness of Forte but also reveal the limitations of relying solely on statistical tests and distribution-level metrics for assessing the similarity between real and synthetic data. We hope that as generative models continue to advance, strong test frameworks like Forte will play a crucial role in maintaining the reliability of ML systems by detecting deviating data, unsafe distributions, distribution shifts, and anomalous samples, ultimately contributing to the development of more robust and trustworthy AI applications in the era of foundation models.

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

# A   APPENDIX: SYNTHETIC DATA SAMPLES

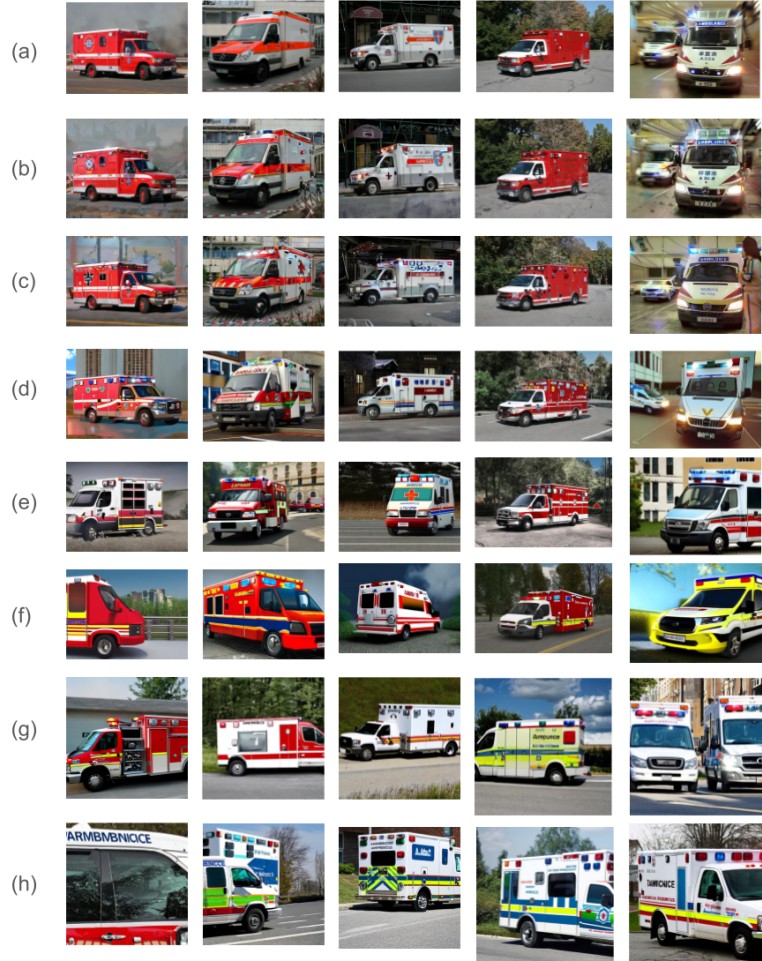

Figure 2: Synthetic images generated for the class "ambulance" using various techniques. Row (a) shows real images, while rows (b) to (f) display Img2Img generated images with strength parameters 0.3, 0.5, 0.7, 0.9, and 1.0, respectively. Row (g) presents images generated using caption descriptions of the real images in row (a), and row (h) shows images generated solely based on the classname.

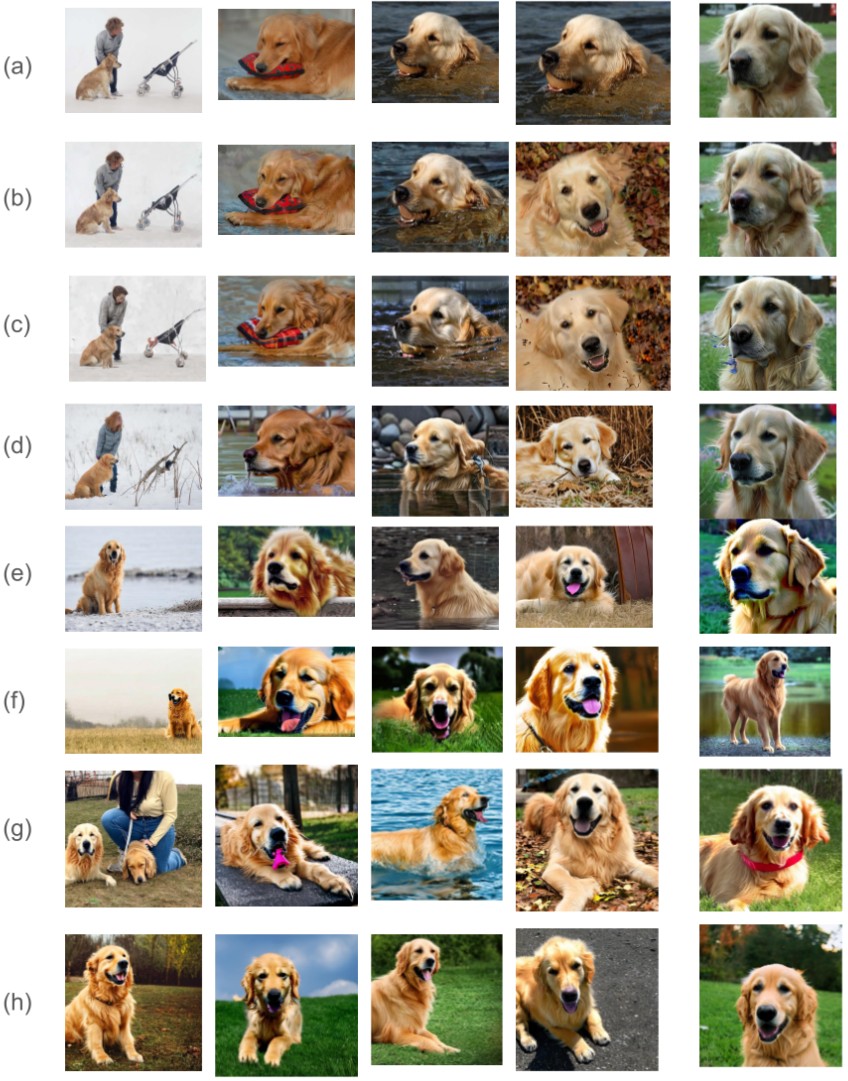

Figure 3: Synthetic images generated for the class "golden retriever" using different techniques. Row (a) shows real images, while rows (b) to (f) display Img2Img generated images with strength parameters 0.3, 0.5, 0.7, 0.9, and 1.0, respectively. Row (g) presents images generated using caption descriptions of the real images in row (a), and row (h) shows images generated solely based on the classname.

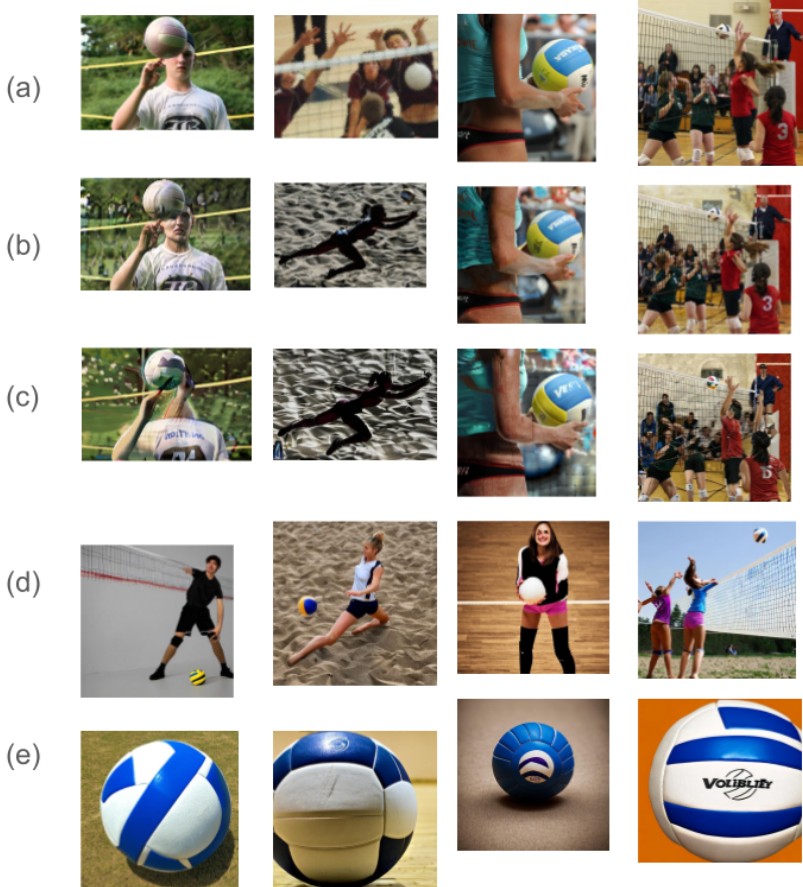

Figure 4: Synthetic images generated for the class "volleyball" using various techniques. Row (a) shows real images, while rows (b) and (c) display Img2Img generated images with strength parameters 0.3, 0.5 respectively. Row (d) presents images generated using caption descriptions of the real images in row (a), and row (e) shows images generated solely based on the classname. Row (e) exhibits mode collapse properties.

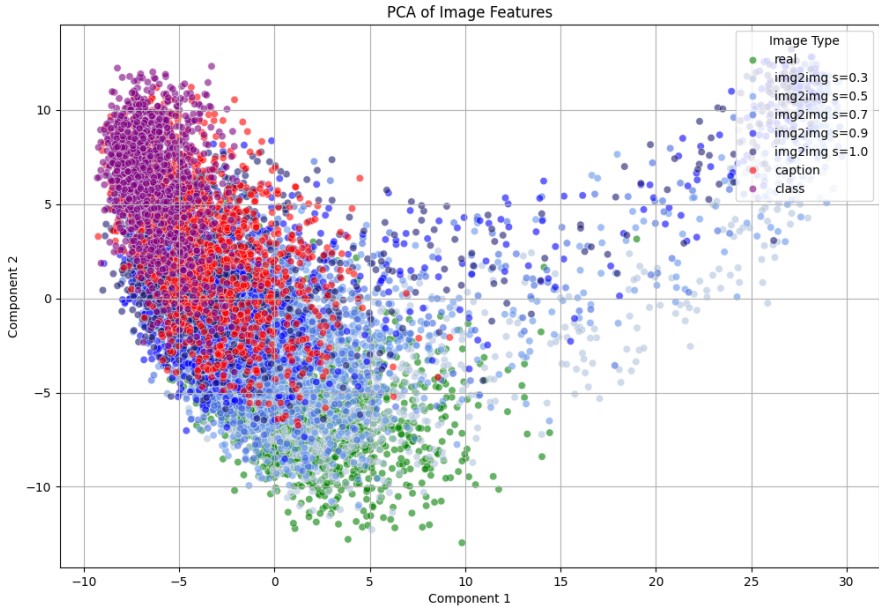

Figure 5: PCA of Golden Retriever Images in CLIP space. We can clearly observe that the distribution of real images is most diverse, whereas the "classname" based generated images are least diverse. We also see a progression of images tending towards the modes of the distribution with increasing strengths allotted to the diffusion process.

# B  APPENDIX: BASELINES STATISTICAL TESTS

Table 6: Comparison of statistical measures for assessing distribution similarity across different image generation methods. The table presents results for various statistical tests and metrics, including Z-score, Kolmogorov-Smirnov (K-S) test, Mann-Whitney U test, Local Outlier Factor (LOF), Isolation Forest (IF), and Wasserstein distance. The image generation methods evaluated include Img2Img with varying strength parameters (S=0.3, 0.5, 0.7, 0.9, 1.0), caption-based generation, and class-based generation for Golden Retrievers. KL Divergence and Jensen Shannon Divergence were calculated on MSN embeddings, since CLIP embeddings had no overlap.

| | Z-Score | K-S Test | | M-W U-Test | | LOF | IF | JSD (MSN) | KLD (MSN) | Wass | Mahalanobis |
|---|---|---|---|---|---|---|---|---|---|---|---|
| | | stat | p-val | stat | p-val | | | | | | |
| Img2Img S=0.3 | -1.17e-9 | 4.17e-3 | 3.30e-4 | 1.27e+11 | 6.64e-1 | 31.11% | 6.11% | 2.75e-3 | 3.47e-3 | 5.52e-3 | 4.38e+1 |
| Img2Img S=0.5 | 4.82e-9 | 4.72e-3 | 2.91e-5 | 2.76e+10 | 2.15e-3 | 27.82% | 5.97% | -1.81e-3 | -2.08e-3 | 6.49e-3 | 4.78e+1 |
| Img2Img S=0.7 | 3.89e-9 | 4.20e-3 | 2.93e-4 | 1.28e+11 | 2.65e-2 | 28.67% | 6.32% | 2.23e-3 | 2.36e-3 | 6.16e-3 | 4.61e+1 |
| Img2Img S=0.9 | -4.16e-9 | 4.43e-3 | 1.09e-4 | 1.27e+11 | 9.20e-2 | 32.61% | 6.92% | 7.22e-3 | 1.02e-2 | 6.57e-3 | 4.22e+1 |
| Img2Img S=1.0 | 7.93e-9 | 4.58e-3 | 5.49e-5 | 1.27e+11 | 3.44e-1 | 34.09% | 7.10% | NaN | NaN | 6.89e-3 | 4.02e+1 |
| Caption-based | 5.56e-9 | 1.02e-2 | 7.56e-23 | 1.29e+11 | 1.85e-29 | 31.83% | 6.38% | NaN | NaN | 1.39e-2 | 4.08e+1 |
| Class-based | -2.98e-9 | 1.17e-2 | 5.19e-30 | 1.29e+11 | 3.26e-40 | 40.63% | 7.73% | NaN | NaN | 1.69e-2 | 3.43e+1 |

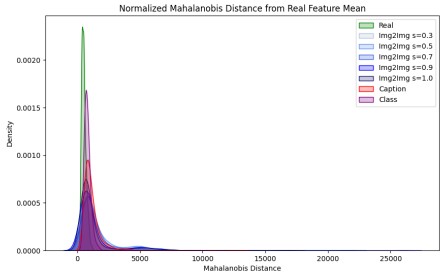

(a) Normalized Mahalanobis Distances for generated distributions vs. real distributions, for the class "golden retriever".

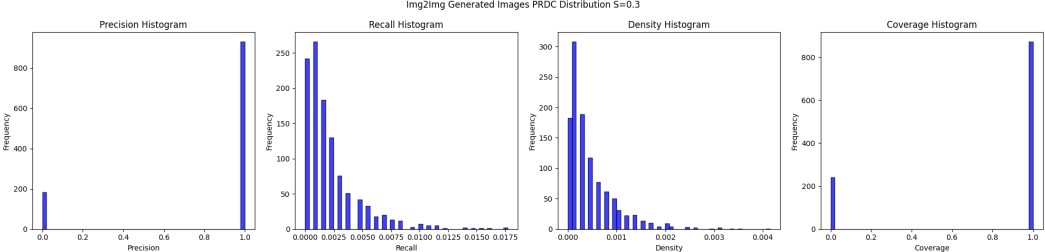

(b) PRDC Value distributions for golden retrievers, with generated images using Stable Diffusion 2 Base with strength = 0.3.

Figure 6: Comparison of Mahalanobis Distances and PRDC Value distributions for real and generated images of golden retrievers.

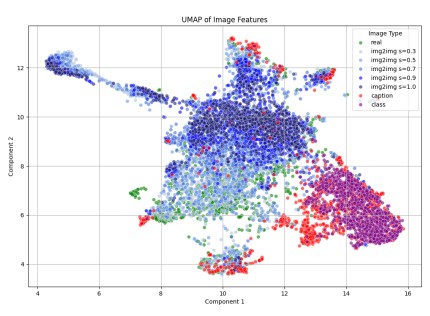

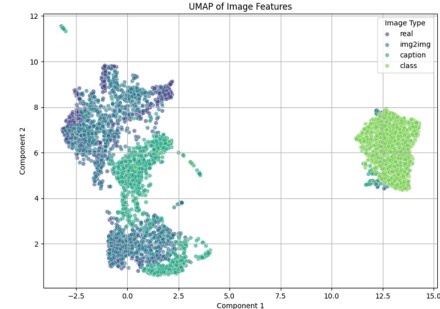

(a) UMAP of Golden Retriever Images in CLIP space.

(b) UMAP of Volleyball Images. Mode collapse is clearly observable for classname generated images.

Figure 7: UMAP visualizations of Golden Retriever and Volleyball images in CLIP space, highlighting mode collapse in classname generated volleyball images.

Table 7: Comparison of statistical measures for assessing distribution similarity across different image generation methods using CLIP embeddings. The table presents results for various statistical tests and metrics, includ- ing Z-score, Kolmogorov-Smirnov (K-S) test, Mann-Whitney U test, Local Outlier Factor (LOF), Isolation Forest (IF), and Wasserstein distance. The image generation methods evaluated include Img2Img with varying strength parameters (S=0.3, 0.5, 0.7, 0.9, 1.0), caption-based generation, and class-based generation for the Volleyball class. KL Divergence and Jensen Shannon Divergence were calculated on MSN embeddings, since CLIP embeddings had no overlap.

| | Z-Score | K-S Test | | M-W U-Test | | LOF | IF | JSD | KLD | Wass | Mahalanobis |
|---|---|---|---|---|---|---|---|---|---|---|---|
| | | stat | p-val | stat | p-val | | | | | | |
| **Img2Img S=0.3** | 2.25e-9 | 4.14e-3 | 3.47e-4 | 1.29e+11 | 1.41e-2 | 27.89% | 4.54% | NaN | NaN | 4.57e-3 | 4.90e+1 |
| **Img2Img S=0.5** | -2.15e-9 | 3.62e-3 | 2.65e-3 | 1.30e+11 | 1.27e-1 | 22.98% | 4.03% | -9.28e-3 | -7.38e-3 | 5.75e-3 | 5.70e+1 |
| **Img2Img S=0.7** | 6.11e-9 | 7.99e-3 | 2.16e-14 | 1.30e+11 | 1.03e-8 | 20.73% | 3.67% | -9.03e-3 | -5.14e-3 | 3.30e-2 | 1.68e+2 |
| **Img2Img S=0.9** | 1.58e-8 | 1.13e-2 | 1.41e-28 | 1.31e+11 | 1.09e-16 | 19.06% | 2.59% | NaN | NaN | 1.12e-2 | 6.66e+1 |
| **Img2Img S=1.0** | 2.06e-9 | 1.22e-2 | 6.56e-33 | 1.31e+11 | 4.99e-17 | 19.42% | 2.26% | NaN | NaN | 1.23e-2 | 6.69e+1 |
| **Caption-based** | 6.77e-9 | 6.91e-3 | 7.12e-11 | 1.30e+11 | 3.46e-10 | 30.89% | 5.27% | NaN | NaN | 7.71e-3 | 4.51e+1 |
| **Class-based** | -5.63e-9 | 2.65e-2 | 2.82e-154 | 1.32e+11 | 2.98e-85 | 36.54% | 2.64% | NaN | NaN | 2.18e-2 | 6.43e+1 |

## C  APPENDIX : THEORETICAL PROPERTIES OF PER-POINT PRDC

**Theorem C.1.** *Under the following assumptions:*

1. *Feature Space: $\mathbb{X} = \mathbb{R}^D$, where $D$ is the dimensionality.*

2. *In-Distribution (ID) Data:*

   - *Training Set: $X_{ID}^{train} = \{\mathbf{x}_1, \ldots, \mathbf{x}_{N_{train}}\}$ with $\mathbf{x}_i \sim \mathcal{N}(\boldsymbol{\mu}_{ID}, \sigma^2 I)$.*
   - *Test Set: $X_{ID}^{test} = \{\mathbf{x}'_1, \ldots, \mathbf{x}'_{N_{test}}\}$ with $\mathbf{x}'_i \sim \mathcal{N}(\boldsymbol{\mu}_{ID}, \sigma^2 I)$.*

3. *Out-of-Distribution (OOD) Data:*

   - *Test Set: $X_{OOD} = \{\mathbf{x}''_1, \ldots, \mathbf{x}''_{N_{OOD}}\}$ with $\mathbf{x}''_i \sim \mathcal{N}(\boldsymbol{\mu}_{OOD}, \sigma^2 I)$, where $\Delta = \|\boldsymbol{\mu}_{ID} - \boldsymbol{\mu}_{OOD}\| \gg 0$.*

4. *Distance Function: Euclidean distance $d(\mathbf{x}, \mathbf{y}) = \|\mathbf{x} - \mathbf{y}\|_2$.*

5. *k-Nearest Neighbors: For a point $\mathbf{x}$, $NN_k(\mathbf{x})$ denotes its k-nearest neighbors in $X_{ID}^{train}$.*

*Using previous definitions of Per-Point PRDC Metrics:*

1. *Precision per point ($P(\mathbf{x}')$):*

$$P(\mathbf{x}') = \mathbb{I}\left[\min_{\mathbf{x} \in X_{ID}^{train}} d(\mathbf{x}', \mathbf{x}) \le r_k(\mathbf{x})\right],$$

   *where $r_k(\mathbf{x})$ is the distance from $\mathbf{x}$ to its k-th nearest neighbor in $X_{ID}^{train}$.*

2. *Recall per point ($R(\mathbf{x}')$):*

$$R(\mathbf{x}') = \frac{1}{N_{train}} \sum_{\mathbf{x} \in X_{ID}^{train}} \mathbb{I}\left[d(\mathbf{x}', \mathbf{x}) \le r_k(\mathbf{x}')\right].$$

3. *Density per point ($D(\mathbf{x}')$):*

$$D(\mathbf{x}') = \frac{1}{k N_{train}} \sum_{\mathbf{x} \in X_{ID}^{train}} \mathbb{I}\left[d(\mathbf{x}', \mathbf{x}) \le r_k(\mathbf{x})\right].$$

4. *Coverage per point ($C(\mathbf{x}')$):*

$$C(\mathbf{x}') = \mathbb{I}\left[\min_{\mathbf{x} \in X_{ID}^{train}} d(\mathbf{x}', \mathbf{x}) \leq r_k(\mathbf{x}')\right],$$

*where $r_k(\mathbf{x}')$ is the distance from $\mathbf{x}'$ to its $k$-th nearest neighbor in $X_{ID}^{train}$.*

*Then, the expected values and variances of these per-point PRDC metrics are:*

- *For ID Test Samples ($\mathbf{x}' \in X_{ID}^{test}$):*

    1. *Expected Precision per point:*
    $$\mathbb{E}[P(\mathbf{x}')] \approx 1 - e^{-k}.$$

    2. *Expected Recall per point:*
    $$\mathbb{E}[R(\mathbf{x}')] \approx \frac{k}{N_{train}}.$$

    3. *Expected Density per point:*
    $$\mathbb{E}[D(\mathbf{x}')] \approx 1.$$

    4. *Expected Coverage per point:*
    $$\mathbb{E}[C(\mathbf{x}')] \approx 1 - e^{-k}.$$

- *For OOD Samples ($\mathbf{x}'' \in X_{OOD}$):*

    1. *Expected Precision, Recall, Density, and Coverage per point are all approximately zero*

- *Variances:*

    1. *Precision and Coverage per point:*
    $$\mathrm{Var}[P(\mathbf{x}')] = \mathbb{E}[P(\mathbf{x}')]\left(1 - \mathbb{E}[P(\mathbf{x}')]\right).$$

    2. *Recall per point:*
    $$\mathrm{Var}[R(\mathbf{x}')] \approx \frac{\mathbb{E}[R(\mathbf{x}')]\left(1 - \mathbb{E}[R(\mathbf{x}')]\right)}{N_{train}}.$$

    3. *Density per point*
    $$\mathrm{Var}[D(\mathbf{x}')] \approx \frac{\mathbb{E}[D(\mathbf{x}')]\left(1 - \mathbb{E}[D(\mathbf{x}')]\right)}{k}.$$

*Proof.* Let us first clarify some assumptions and notation, specific to this proof for clarity.

1. Feature Space: $\mathbb{X} = \mathbb{R}^D$.

2. ID Data Distribution: $\mathcal{N}(\boldsymbol{\mu}_{\mathrm{ID}}, \sigma^2 I)$.

3. OOD Data Distribution: $\mathcal{N}(\boldsymbol{\mu}_{\mathrm{OOD}}, \sigma^2 I)$, with the mean difference $\Delta = \|\boldsymbol{\mu}_{\mathrm{ID}} - \boldsymbol{\mu}_{\mathrm{OOD}}\| \gg 0$.

4. Sample Sizes:

    (a) $N_{\mathrm{train}}$: Number of ID training samples.
    (b) $N_{\mathrm{test}}$: Number of ID test samples.
    (c) $N_{\mathrm{OOD}}$: Number of OOD samples.

5. Distance Function: $d(\mathbf{x}, \mathbf{y}) = \|\mathbf{x} - \mathbf{y}\|_2$.

6. $k$-Nearest Neighbor Distances:

    (a) $r_k(\mathbf{x})$: Distance from $\mathbf{x}$ to its $k$-th nearest neighbor in $X_{\mathrm{ID}}^{\mathrm{train}}$.
    (b) $r_k(\mathbf{x}')$: Distance from $\mathbf{x}'$ to its $k$-th nearest neighbor in $X_{\mathrm{ID}}^{\mathrm{train}}$.

**Calculating Distribution of Distances**

**Distance Between ID Samples**

Let $\mathbf{x}, \mathbf{y} \sim \mathcal{N}(\boldsymbol{\mu}_{\text{ID}}, \sigma^2 I)$. The difference $\mathbf{x} - \mathbf{y} \sim \mathcal{N}(\mathbf{0}, 2\sigma^2 I)$.

The squared distance $d^2(\mathbf{x}, \mathbf{y})$ is the sum of $D$ squared normal variables :

$$d^2(\mathbf{x}, \mathbf{y}) = \sum_{i=1}^{D} (x_i - y_i)^2.$$

where, each $(x_i - y_i)$ is $\sim \mathcal{N}(0, 2\sigma^2)$. Therefore, $\frac{d^2(\mathbf{x},\mathbf{y})}{2\sigma^2} \sim \chi_D^2$, i.e. follows a chi-squared distribution with $D$ degrees of freedom.

Expected and Variance of the Squared Distance:

$$\mathbb{E}[d^2(\mathbf{x}, \mathbf{y})] = 2\sigma^2 D.$$

$$\text{Var}[d^2(\mathbf{x}, \mathbf{y})] = (2\sigma^2)^2 \cdot 2D = 4\sigma^4(2D) = 8\sigma^4 D.$$

**Distance Between ID and OOD Samples**

Let $\mathbf{x} \sim \mathcal{N}(\boldsymbol{\mu}_{\text{ID}}, \sigma^2 I)$ and $\mathbf{x}'' \sim \mathcal{N}(\boldsymbol{\mu}_{\text{OOD}}, \sigma^2 I)$.

- The difference $\mathbf{x} - \mathbf{x}'' \sim \mathcal{N}(\boldsymbol{\delta}, 2\sigma^2 I)$, where $\boldsymbol{\delta} = \boldsymbol{\mu}_{\text{ID}} - \boldsymbol{\mu}_{\text{OOD}}$.

- The squared distance follows a non-central chi-squared distribution:

$$\frac{d^2(\mathbf{x}, \mathbf{x}'')}{2\sigma^2} \sim \chi_D^2(\lambda),$$

where $\lambda = \frac{\|\boldsymbol{\delta}\|^2}{2\sigma^2}$.

- Therefore, the Expected Squared Distance of a non-central chi-squared distribution is:

$$\mathbb{E}[d^2(\mathbf{x}, \mathbf{x}'')] = 2\sigma^2(D + \lambda) = 2\sigma^2 D + \Delta^2.$$

- And the Variance of Squared Distance of a non-central chi-squared distribution is

$$\text{Var}[d^2(\mathbf{x}, \mathbf{x}'')] = 4\sigma^4(2D + 4\lambda) = 8\sigma^4 D + 16\sigma^4 \lambda.$$

**Expected $k$-Nearest Neighbor Distance $r_k(\mathbf{x})$**

- Distances from $\mathbf{x}$ to other training samples are i.i.d. and follow the distribution of $d(\mathbf{x}, \mathbf{y})$ as previously defined.

- Empirical CDF of Distances:

$$F(d) = P(d(\mathbf{x}, \mathbf{y}) \leq d).$$

- Expected $k$-th Nearest Neighbor Distance:

    1. For large $D$ (number of dimensions in the vector), the chi-squared distribution can be approximated by a normal distribution:

    $$\chi_D^2 \approx \mathcal{N}(D, 2D).$$

    2. This yields standard normal statistic

    $$\frac{\frac{d^2}{2\sigma^2} - D}{\sqrt{2D}} \sim \mathcal{N}(0, 1)$$

3. Since p-value of a test statistic follows a standard uniform distribution under the null hypothesis:

$$\Phi\left(\frac{\frac{d^2}{2\sigma^2} - D}{\sqrt{2D}}\right) \sim u(0, 1)$$

4. If $f$ is a monotonically increasing function of variable $X$ whose $k$th order statistic in a sample of size $n$ is $X_{(k)}$, then the $k$th order statistic of $f(X)$ is $f(X_{(k)})$ when $X$ is taken from the sample. Note that $\Phi$ is monotonically increasing as it is a CDF, and so is $f(X) = \frac{\frac{X^2}{2\sigma^2} - D}{\sqrt{2D}}$ on the positive reals which are the domain of $r_k(\mathbf{x})$.

5. By definition $d_{(k)} = r_k(\mathbf{x})$ for some $\mathbf{x}$. Then if $U = \Phi\left(\frac{\frac{d^2}{2\sigma^2} - D}{\sqrt{2D}}\right)$ is a standard uniform, its $k$th order statistic in the sample is

$$U_{(k)} = \Phi\left(\frac{\frac{(r_k(\mathbf{x}))^2}{2\sigma^2} - D}{\sqrt{2D}}\right)$$

6. The $k$th order statistic of a standard uniform follows a Beta distribution with $\alpha = k$ and $\beta = n - k + 1$:

$$\Phi\left(\frac{\frac{(r_k(\mathbf{x}))^2}{2\sigma^2} - D}{\sqrt{2D}}\right) \sim \text{Beta}(k, n - k + 1)$$

7. Because the expected value of a Beta distribution is $\frac{\alpha}{\alpha+\beta}$:

$$\mathbb{E}\left[\Phi\left(\frac{\frac{(r_k(\mathbf{x}))^2}{2\sigma^2} - D}{\sqrt{2D}}\right)\right] = \frac{k}{n + 1}$$

**Expected PRDC Metrics for ID Samples**

**Expected Precision per point** ($\mathbb{E}[\text{P}(\mathbf{x}')]$)

- Definition:
$$\mathbb{E}[\text{P}(\mathbf{x}')] = \mathbb{E}[\min_{\mathbf{x} \in \mathbf{X}_{\text{ID}}^{\text{train}}} P\left(d(\mathbf{x}', \mathbf{x}) \le r_k(\mathbf{x})\right)] = 1 - \prod_{\mathbf{x} \in X_{\text{ID}}^{\text{train}}} \left(1 - \mathbb{E}[P\left(d(\mathbf{x}', \mathbf{x}) \le r_k(\mathbf{x})\right)]\right).$$

- Also:
$$P\left(d(\mathbf{x}', \mathbf{x}) \le r_k(\mathbf{x})\right)] = P\left(\frac{\frac{d(\mathbf{x}',\mathbf{x})^2}{2\sigma^2} - D}{\sqrt{2D}} \le \frac{\frac{(r_k(\mathbf{x}))^2}{2\sigma^2} - D}{\sqrt{2D}}\right) = \Phi\left(\frac{\frac{(r_k(\mathbf{x}))^2}{2\sigma^2} - D}{\sqrt{2D}}\right)$$

- Probability for a Single Training Sample, using the above Beta distribution formula, is
$$\mathbb{E}\left[P\left(d(\mathbf{x}', \mathbf{x}) \le r_k(\mathbf{x})\right)\right] = \mathbb{E}\left[\Phi\left(\frac{\frac{r_k^2(x)}{2\sigma^2} - D}{\sqrt{2D}}\right)\right] = \frac{k}{N_{\text{train}} + 1}$$

- Expected Precision per point:

  Because we are dealing with in-distribution data and p-values are uniformly distributed under the null hypothesis,
  $$\mathbb{E}[\text{P}(\mathbf{x}')] = 1 - \left(1 - \frac{k}{N_{\text{train}} + 1}\right)^{N_{\text{train}}} \approx 1 - e^{-k}.$$
  since,
  $$\lim_{n\to\infty}\left(1 - \frac{k}{n+1}\right)^n = \lim_{n\to\infty}\frac{\left(1 - \frac{k}{n+1}\right)^{n+1}}{1 - \frac{k}{n+1}} = \frac{e^{-k}}{1 - 0} = e^{-k}$$

**Expected Recall per point ($\mathbb{E}[\mathbf{R}(\mathbf{x}')]$)**

- Definition:

$$\mathbf{R}(\mathbf{x}') = \frac{1}{N_{\text{train}}} \sum_{\mathbf{x} \in X_{\text{ID}}^{\text{train}}} \mathbb{I}\left[d(\mathbf{x}', \mathbf{x}) \le r_k(\mathbf{x}')\right].$$

- Understanding $r_k(\mathbf{x}')$: For each $\mathbf{x}'$, $r_k(\mathbf{x}')$ is the distance to its $k$-th nearest neighbor among the $N_{\text{train}}$ training samples. The distances $d_i = d(\mathbf{x}', \mathbf{x}_i)$ for $i = 1, \ldots, N_{\text{train}}$ are independent and identically distributed (i.i.d.) random variables because $\mathbf{x}'$ and $\mathbf{x}_i$ are independent samples from the same distribution.

- Probability that a Training Sample is within $r_k(\mathbf{x}')$: The variable $\mathbb{I}\left[d(\mathbf{x}', \mathbf{x}_i) \le r_k(\mathbf{x}')\right]$ is an indicator that is 1 if $\mathbf{x}_i$ is among the $k$ nearest neighbors of $\mathbf{x}'$ in the training set. The probability that a particular training sample $\mathbf{x}_i$ is within the $k$-nearest neighbors of $\mathbf{x}'$ is:

$$P\left(d(\mathbf{x}', \mathbf{x}_i) \le r_k(\mathbf{x}')\right) = \frac{k}{N_{\text{train}}}$$

This is since the $k$ nearest neighbors are chosen among $N_{\text{train}}$ samples, each training sample has an equal chance of $\frac{k}{N_{\text{train}}}$ of being among the closest $k$ samples to $\mathbf{x}'$.

$$\mathbb{E}\left[\mathbb{I}\left[d(\mathbf{x}', \mathbf{x}_i) \le r_k(\mathbf{x}')\right]\right] = \frac{k}{N_{\text{train}}}$$

$$\mathbb{E}[\mathbf{R}(\mathbf{x}')] = \frac{1}{N_{\text{train}}} \sum_{i=1}^{N_{\text{train}}} \mathbb{E}\left[\mathbb{I}\left[d(\mathbf{x}', \mathbf{x}_i) \le r_k(\mathbf{x}')\right]\right] = \frac{1}{N_{\text{train}}} \times N_{\text{train}} \times \frac{k}{N_{\text{train}}} = \frac{k}{N_{\text{train}}}$$

$$\mathbb{E}[\mathbf{R}(\mathbf{x}')] = \frac{k}{N_{\text{train}}}$$

**Expected Density per point ($\mathbb{E}[\mathbf{D}(\mathbf{x}')]$)**

- Definition:

$$\mathbf{D}(\mathbf{x}') = \frac{1}{k} \sum_{\mathbf{x} \in X_{\text{ID}}^{\text{train}}} \mathbb{I}\left[d(\mathbf{x}', \mathbf{x}) \le r_k(\mathbf{x})\right].$$

- Expected Sum, for large N:

$$\mathbb{E}\left[\sum_{\mathbf{x} \in X_{\text{ID}}^{\text{train}}} \mathbb{I}\left[d(\mathbf{x}', \mathbf{x}) \le r_k(\mathbf{x})\right]\right] = N_{\text{train}} \times \frac{k}{N_{\text{train}} + 1} \approx k.$$

- Expected Density per point:

$$\mathbb{E}[\mathbf{D}(\mathbf{x}')] \approx \frac{k}{k} = 1.$$

**Expected Coverage per point ($\mathbb{E}[\mathbf{C}(\mathbf{x}')]$)**

- Given both $\mathbf{x}'$ and the training samples $\mathbf{x}$ are drawn from the same distribution $\mathcal{N}(\boldsymbol{\mu}_{\text{ID}}, \sigma^2 I)$.

- The minimum distance $\min_{\mathbf{x}_i} d(\mathbf{x}', \mathbf{x}_i)$ is the smallest among $N_{\text{train}}$ distances drawn from the chi-squared distribution.

- Distance to $k$-th Nearest Neighbor $r_k(\mathbf{x}')$ is $r_k(\mathbf{x}')$ is the $k$-th order statistic of the distances $d(\mathbf{x}', \mathbf{x}_i)$. We are interested in the probability:

$$P\left(\min_{\mathbf{x}_i} d(\mathbf{x}', \mathbf{x}_i) \le r_k(\mathbf{x}')\right)$$

- Since the minimum distance is always less than or equal to the $k$-th smallest distance, the event is always true:

$$\min_{\mathbf{x}_i} d(\mathbf{x}', \mathbf{x}_i) \leq r_k(\mathbf{x}')$$

Therefore:

$$P\left(\min_{\mathbf{x}_i} d(\mathbf{x}', \mathbf{x}_i) \leq r_k(\mathbf{x}')\right) = 1$$

This is because the closest training sample to $\mathbf{x}'$ is by definition among the $k$-nearest neighbors used to compute $r_k(\mathbf{x}')$. Therefore:

$$\mathbb{E}[C(\mathbf{x}')] = 1$$

**Expected PRDC Metrics for OOD Samples**

- Due to the large distance $\Delta$, the probability that an OOD sample is within $r_k(\mathbf{x})$ of any ID training sample is negligible.

- Therefore:

  1. $\mathbb{E}[P(\mathbf{x}'')] \approx 0$.
  2. $\mathbb{E}[R(\mathbf{x}'')] \approx 0$.
  3. $\mathbb{E}[D(\mathbf{x}'')] \approx 0$.
  4. $\mathbb{E}[C(\mathbf{x}'')] \approx 0$.

**Variances of PRDC Metrics**

**Variance of Precision and Coverage per point**

1. Since these are Bernoulli random variables:

$$\mathrm{Var}[P(\mathbf{x}')] = \mathbb{E}[P(\mathbf{x}')]\left(1 - \mathbb{E}[P(\mathbf{x}')]\right) = e^{-k} - e^{-2k}.$$

2. The variance of coverage per point for ID samples is zero. This reflects the certainty that ID test samples will always satisfy the coverage condition.

**Variance of Recall per point**

1. Recall per point is an average of $N_{\text{train}}$ Bernoulli variables with success probability

$$p = \frac{k}{N_{\text{train}}}$$

.

2. Because the sum of $N_{\text{train}}$ Bernoulli variables with probability $p$ is a binomial distribution and we are dividing that sum by $N_{\text{train}}$, the Variance becomes:

$$\mathrm{Var}[R(\mathbf{x}')] = \frac{1}{N_{\text{train}}^2} \times N_{\text{train}} \times p\left(1 - p\right) = \frac{p(1 - p)}{N_{\text{train}}}.$$

**Variance of Density per point**

- Density per point sums $N_{\text{train}}$ Bernoulli variables and divides by $k$.

- Variance:

$$\mathrm{Var}[D(\mathbf{x}')] = \frac{1}{k^2} \times N_{\text{train}} \times p\left(1 - p\right) = \frac{p(1 - p)N_{\text{train}}}{k^2}.$$

- Since $p = \frac{k}{N_{\text{train}}+1}$, we get:

$$\mathrm{Var}[D(\mathbf{x}')] \approx \frac{1}{k}\left(1 - \frac{k}{N_{\text{train}}}\right).$$

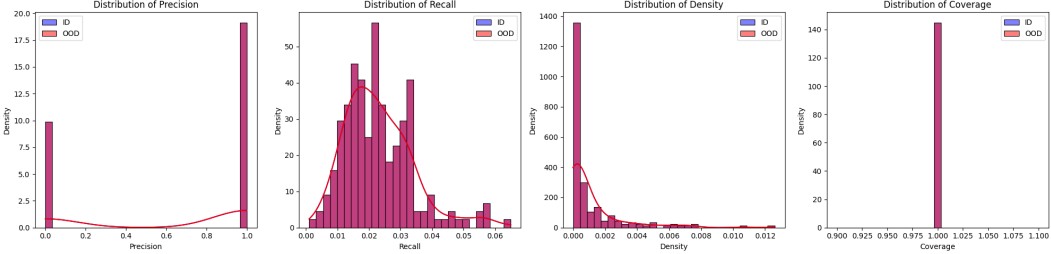

Figure 8: Distribution of PRDC metrics for the scenario where both the training (ID) and test (ID) data are drawn from the same Gaussian distribution with zero mean. Blue lines represent the ID data, and red lines represent the OOD data (which is actually the same as ID in this case). The histograms overlap entirely, indicating identical distributions for all metrics due to the same underlying data.

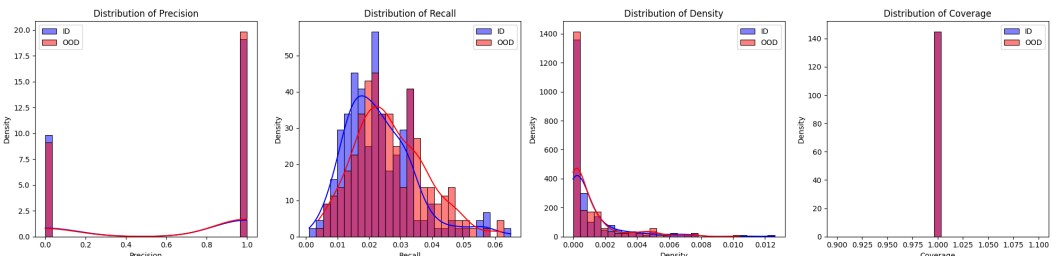

Figure 9: Distribution of PRDC metrics for the scenario with a moderate shift, where the OOD test data is generated from a Gaussian distribution with its mean shifted by 0.01 units compared to the ID data. Blue lines represent the ID data, and red lines represent the moderately shifted OOD data. The histograms show noticeable differences and partial overlap between ID and OOD distributions, reflecting the moderate shift in data.

- For large $N_{\text{train}}$, this simplifies to:

$$\text{Var}[\mathbf{D}(\mathbf{x}')] \approx \frac{1}{k}.$$

- For ID samples, per-point Precision and Coverage have expected values close to $1 - e^{-k}$, which increases with $k$.

- Per-point Recall is small when $k \ll N_{\text{train}}$, with $\mathbb{E}[\mathbf{R}(\mathbf{x}')] \approx \frac{k}{N_{\text{train}}}$.

- Per-point Density is approximately 1 for ID samples, regardless of $k$.

- For OOD samples, all per-point PRDC metrics are approximately zero due to the large mean difference $\Delta$.

- Variances depend on $k$ and $N_{\text{train}}$, but are generally small for large datasets.

$\square$

**Note on Assumptions and Approximations**

- Independence Assumption: While distances are not strictly independent, the independence approximation simplifies calculations and is reasonable for large datasets.

- Large $D$ Approximation: Approximations using the normal distribution for the chi-squared distribution are valid when $D$ is large.

- Small $k$ Relative to $N_{\text{train}}$: The results are most accurate when $k \ll N_{\text{train}}$.

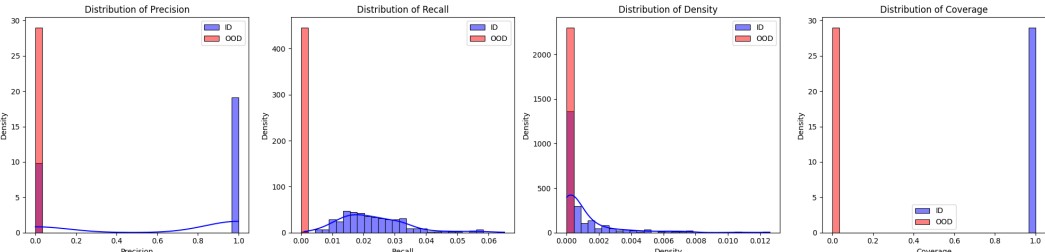

Figure 10: Distribution of PRDC metrics for the scenario with a large shift, where the OOD test data is generated from a Gaussian distribution with its mean shifted by 5 units from the ID data. Blue lines represent the ID data, and red lines represent the significantly shifted OOD data. The histograms display minimal overlap, indicating that the PRDC metrics effectively distinguish between the ID and heavily shifted OOD data.

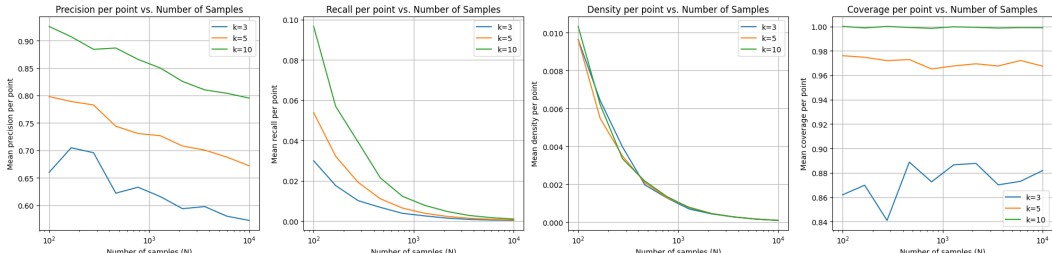

Figure 11: Line plots showing the mean PRDC metrics—Precision, Recall, Density, and Coverage—per data point as a function of the number of samples (N) for varying values of k (number of nearest neighbors). Each subplot corresponds to one of the PRDC metrics, with different lines representing k=3, k=5, and k=10. The semi-logarithmic x-axis emphasizes the impact of sample size on the stability and reliability of the PRDC metrics across different neighborhood sizes. Notice that in precision and coverage, the variance goes down considerably (theoretically, exponentially) on the basis of k, supported by our theorem.

## C.1 ROBUSTNESS TO THE CURSE OF DIMENSIONALITY

**Intuition:** The curse of dimensionality often manifests as the concentration of distances in high-dimensional spaces. By reducing to four scalar metrics, PRDC sidesteps many of these issues. Let $X_n$ be uniformly distributed in the $n$-dimensional unit ball. As $n \to \infty$: $\frac{\max_{1 \le i < j \le k} |X_i - X_j|}{\min_{1 \le i < j \le k} |X_i - X_j|} \xrightarrow{P} 1$ for any fixed $k$. As the number of dimensions n increases, the ratio of the maximum to minimum distances between any pair of k points approaches 1 in probability. This means that the distances between points become almost the same, regardless of their specific locations in the unit ball. This contributes to why directly using high dimensional representations does not help.

PRDC metrics are not directly affected by this distance concentration, as they capture relative rather than absolute distances. These metrics provide more meaningful comparisons between datasets, especially in high-dimensional spaces where traditional distance measures lose their discriminatory power. By emphasizing relative rather than absolute distances, PRDC metrics are less affected by the concentration of distances in high-dimensional spaces.

**Experiment:** We want to understand how increasing the dimensionality of data affects the ability of PRDC metrics and other measures to distinguish between inliers and outliers.

We use the following methodology for data generation :

- Inliers: For each dimension $D$ ranging from 2 to 200 (in steps of 5), we generate 1000 samples from a $D$-dimensional standard normal distribution centered at the origin:

$$\mathbf{x}_{\text{inlier}} \sim \mathcal{N}(\mathbf{0}, I_D),$$

  where $I_D$ is the $D \times D$ identity matrix.

- We generate 100 samples from a $D$-dimensional normal distribution centered at $\boldsymbol{\mu} = [3, 3, \ldots, 3]$:

$$\mathbf{x}_{\text{outlier}} \sim \mathcal{N}(3\mathbf{1}, I_D),$$

  where $\mathbf{1}$ is a vector of ones.

We calculate the PRDC per point metrics, and then the following additional measures :

- Mean Euclidean Distance:

$$\text{Mean Distance} = \frac{1}{N} \sum_{i=1}^{N} \|\mathbf{x}_i\|_2,$$

  where $N$ is the number of inliers.

- Mean Cosine Similarity:

$$\text{Mean Cosine Similarity} = \frac{2}{N(N-1)} \sum_{i<j} \frac{\mathbf{x}_i^\top \mathbf{x}_j}{\|\mathbf{x}_i\|_2 \|\mathbf{x}_j\|_2}.$$

- PCA Variance Explained: Sum of variance explained by the first two principal components obtained from Principal Component Analysis (PCA) on the inliers.

We iterate over each dimension $D$ and:

1. Generate inliers and outliers.
2. Compute PRDC metrics for inliers vs. inliers and inliers vs. outliers.
3. Compute additional measures.
4. Record the results for analysis.

**Observations**

- **Precision:** Inliers exhibit high precision across dimensions, indicating that inliers are close to each other. For outliers, precision decreases with increasing $D$, and then stabilizes, due to increased distances in high-dimensional spaces.

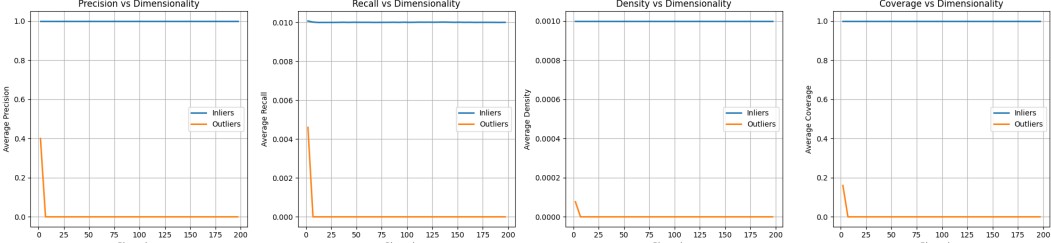

Figure 12: Precision, Recall, Density, and Coverage (PRDC) metrics plotted against the number of dimensions for Inlier (ID) and Outlier (OOD) datasets. In each subplot, the blue line represents the average PRDC metric for inliers compared against themselves, while the orange line depicts the average PRDC metric for outliers compared against inliers. This figure illustrates how increasing dimensionality impacts the effectiveness of PRDC metrics in distinguishing between inliers and outliers across different degrees of distributional shift.

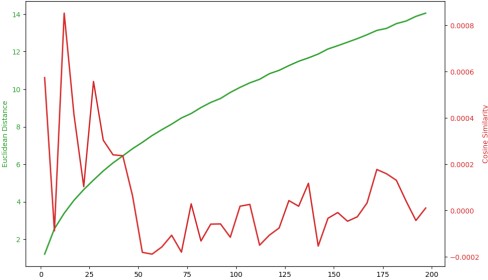

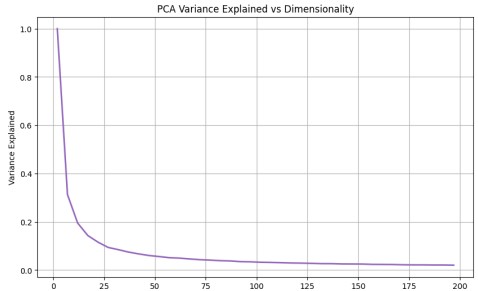

(a) Average Euclidean Distance from the Origin and Average Cosine Similarity among Inliers plotted against the number of dimensions for Inlier (ID) datasets. The left y-axis (colored in green) shows the mean Euclidean distance of inliers from the origin, while the right y-axis (colored in orange) displays the mean cosine similarity among inliers. This dual-axis plot highlights how higher dimensionality leads to increased Euclidean distances and decreased cosine similarities, reflecting the effects of the curse of dimensionality on data geometry.

(b) Total variance explained by the first two Principal Components (PC1 and PC2) of Inlier (ID) datasets plotted against the number of dimensions. The blue line represents the cumulative variance captured by the first two principal components. As dimensionality increases, the proportion of variance explained by the first two components decreases, indicating that data variance becomes more dispersed across additional dimensions. This trend underscores the challenges of dimensionality reduction in high-dimensional spaces, sidestepped by Forte.

- **Recall:** Inliers show higher recall, reflecting better coverage among inliers. Outliers on the other hand show lower recall, as outliers do not cover the inlier distribution well.

- **Density:** Inliers demonstrate high density, indicating dense clustering. For outliers, density decreases with $D$, showing sparse connections to inliers.

- **Coverage:** Inliers exhibit high coverage, demonstrating that they effectively cover the inlier distribution. Outlier coverage decreases with $D$, indicating poor coverage of inliers by outliers.

- **Mean Euclidean Distance:** This metric increases with $D$ for both inliers and outliers, due to the phenomenon where distances between points in high-dimensional spaces tend to increase.

- **Mean Cosine Similarity:** This measure decreases with $D$, approaching zero for both inliers and outliers, as random vectors in high dimensions become nearly orthogonal.

- **PCA Variance Explained:** This metric decreases with $D$ for both inliers and outliers, as variance is distributed among more components, making it harder to capture significant variance in the first few components.

## C.2 PRDC AS LOCAL TRANSFORMATIONS

A local transformation is a function applied to a vector in a space which depends not only on the vector itself but also on its neighboring vectors. This allows for contextual dependence, where the transformation depends on the local environment of the vector, therefore making it sensitive to local variations and patterns. This also allows the local transformations to be non-linear, allowing for complex manipulations that are not possible with global linear transformations. This means they can adapt to regions of the data space, capturing variations and features that might be missed by global transformations.

These are obviously very powerful, as evidenced by convolution kernels (depending on neighbour pixels), Graph convolutions, Laplacian smoothing, and wavelet transforms. However, they can be sensitive to parameters like kernel size, weights, and tuning is important to ensure robustness. PRDC metrics might look very different at different k values, for example.

## C.3 KEY PROPERTIES AND CHARACTERISTICS OF THE FORTE ALGORITHM

At its core, Forte leverages locality, basing its per-point PRDC (Precision, Recall, Density, Coverage) metrics on local neighborhood structures. This approach makes the algorithm sensitive to local variations in data distribution. Simultaneously, Forte performs dimensionality reduction, compressing high-dimensional data ($D \gg 4$) into a more manageable $\mathbb{R}^4$ space while preserving essential information for OOD detection. This reduction not only aids in mitigating overfitting but also improves generalization and computational efficiency.

A key strength of Forte lies in its model-agnostic nature, allowing it to work with any feature extractor that provides meaningful representations in $\mathbb{R}^D$. Self-supervised models like CLIP, ViTMSN, and DINOv2 are particularly effective in this context due to their rich feature representations. The algorithm's non-parametric approach, which avoids assumptions about the data distribution's form, contributes to its flexibility and robustness. This is further enhanced by the tunable parameter $k$ (number of nearest neighbors), which allows for balancing sensitivity to local structures with noise robustness.

Forte's effectiveness in OOD detection stems from its ability to capture both density (proximity of neighbors) and coverage (sample's position within the support of $P_{\text{ID}}$). By focusing on local neighborhood information, the algorithm can detect subtle discrepancies between in-distribution and OOD samples, particularly effective when OOD samples reside in low-density regions. The use of multiple metrics (precision, recall, density, and coverage) provides a holistic view of how each sample relates to the training data $X_{\text{ID}}^{\text{train}}$, enhancing the algorithm's discriminative power. The local transform employed by Forte also confers robustness to feature space variability, an important consideration when working with different self-supervised learning models. By normalizing differences through a focus on relative distances within the feature space, Forte maintains consistency across varied feature representations. Empirical results presented in this paper have comprehensively demonstrated Forte's

superior performance compared to traditional methods, showcasing how the combination of powerful feature extractors and the local transform leads to effective OOD detection.

## D   APPENDIX : ABLATION STUDY ON ENCODERS

**Insight into encoders:** The results presented in Table. 3 demonstrate that richer representations significantly improve OOD detection performance, as evidenced by the ranking of encoders: CLIP > DINO v2 > MSN when used individually, and CLIP + DINO v2 > CLIP + MSN > DINO v2 + MSN in the two-model combinations. To further investigate this phenomenon, we conducted additional experiments. Specifically, we included DeIT with both ViT-B (Base) and ViT-Ti (Tiny) models and evaluated their OOD detection performance under the settings studied in Tables 2. These results, show that DeIT-B achieves 0.87 AUROC on CIFAR-100, while DeIT-Ti achieves 0.82 AUROC. This aligns with our hypothesis that more informative representations are essential for effective OOD detection. DeIT, trained with an objective approximating supervision, produces less informative embeddings compared to self-supervised encoders like DINO v2. Similarly, the DeIT-Ti model performs worse due to its reduced capacity for generating robust representations. We think these findings provide valuable insights into the utility of different encoders for OOD detection and offer guidance for practitioners seeking optimal performance.

Table 8: Comparison of AUROC and FPR95 performance figures for Base and Tiny DeIT models across the tasks in

| Model | In-Dist | OOD Dataset | AUROC | FPR95 |
|-------|---------|-------------|-------|-------|
| Base-DeIT | CIFAR-10 | CIFAR-100 | 0.8712 | 0.9926 |
| Tiny-DeIT | CIFAR-10 | CIFAR-100 | 0.8261 | 0.9903 |
| Base-DeIT | CIFAR-10 | SVHN | 0.9554 | 0.4604 |
| Tiny-DeIT | CIFAR-10 | SVHN | 0.9296 | 0.6195 |
| Base-DeIT | CIFAR-10 | Celeb-A | 0.9871 | 0.0015 |
| Tiny-DeIT | CIFAR-10 | Celeb-A | 0.9929 | 0.0007 |

## E   APPENDIX: MEDICAL IMAGE DATASETS

In medical imaging research, studies are often done using one in-house dataset. Conclusions and models drawn from these studies are then applied to new data, with poor results. In particular, MRI datasets exhibit strong batch effects that prevent them from being in-distribution relative to each other because some acquisition protocol is bound to be different between them. Moreover, dataset sizes are severely limited in clinical applications, which means a separate model cannot be trained for each batch. A single model cannot be robust to all MRI datasets of the same subject matter, even if they have similar acquisition parameters. Such datasets still carry enough differences to impact model performance.

To simulate this scenario, two public datasets are used for the experiments in Section 5.2: coronal knee MRI from FastMRI Zbontar et al. (2018); Knoll et al. (2020) with two subsets and Osteoarthritis Initiative (OAI) Nevitt et al. (2006) with three subsets. The acquisition parameters, including sequence and fat suppression are detailed in Table 9 and samples are shown in Figure 14. Treating the FastMRI dataset as in-distribution and assuming that models have been trained on them, Forte is used to determine the next course of action when confronted with the OAI dataset: 1) which subsets of the new dataset can be aligned with the existing subsets / models? 2) To what degree do these subsets diverge from each other? Using the FastMRI FS subset as in-distribution, the two OAI subsets (OAI T1 and OAI MPR) are tested for OOD detection. Similarly, the FastMRI NoFS subset is used as in-distribution and the OAI TSE subset is tested for OOD detection.

Table 9: Acquisition parameters for MRI, grouped by distributions as used in Section 5.2.

| Parameter | FastMRI NoFS | FastMRI FS | OAI TSE | OAI T1 | OAI MPR |
|---|---|---|---|---|---|
| Sequence | 2D TSE PD | 2D TSE PD | 2D TSE T1w | 3D FLASH T1w | 3D DESS T2w |
| FOV (mm$^2$) | 140 × 140 | 140 × 140 | 140 × 140 | 140 × 140 | 140 × 140 |
| Matrix size | 320 × 320 | 320 × 320 | 320 × 320 | 384 × 384 | 320 × 320 |
| Slice thickness (mm) | 3 | 3 | 3 | 0.7 | 1.5 |
| TR (ms) | 2750–3000 | 2850–3000 | 800 | 9.7 | 14.7 |
| TE (ms) | 27–32 | 33 | 9 | 4.0 | 4.2 |
| Fat suppression | No | Yes | No | Yes | Yes |

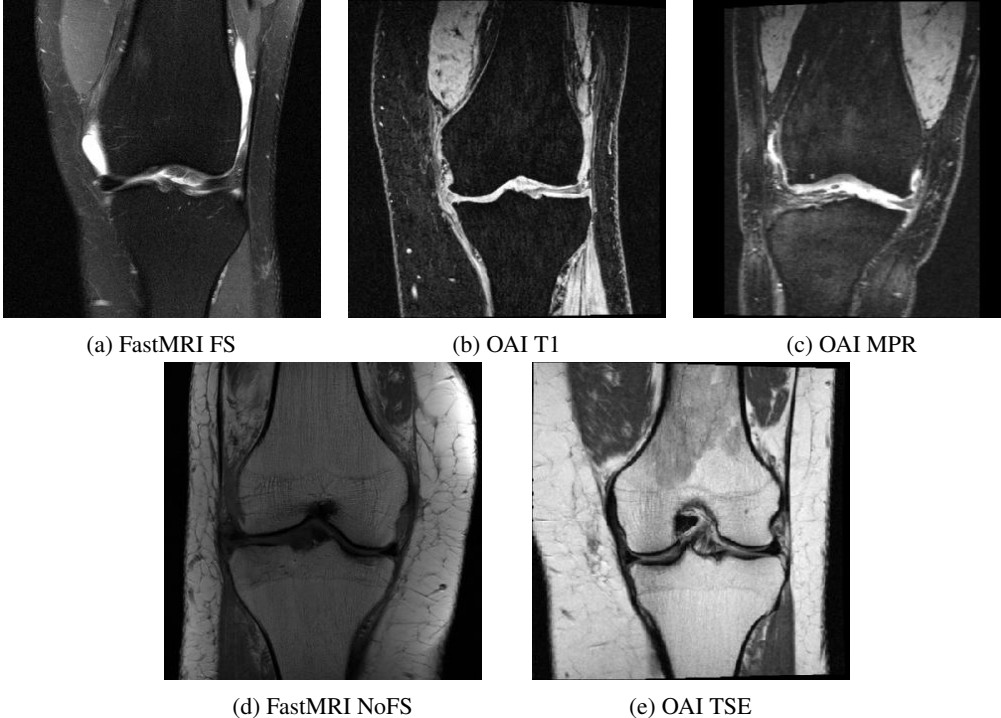

(a) FastMRI FS     (b) OAI T1     (c) OAI MPR

(d) FastMRI NoFS     (e) OAI TSE

Figure 14: Sample images from five total subsets of the OAI and FastMRI datasets. Fat-suppressed (a-c) and non fat-suppressed (d,e) subsets tested for OOD.

# F  PSEUDOCODE

---

**Algorithm 1** OOD Detection Using Per-Point PRDC Metrics in Forte

---

**Input:** Reference data features $\{x_j^r\}_{j=1}^m$
**Input:** Test data features $\{x_i^g\}_{i=1}^n$
**Input:** Number of nearest neighbors $k$
**Output:** OOD detection performance metrics: AUROC, FPR@95
1: **Feature Extraction (Preprocessing)**:
   Use pre-trained models (e.g., CLIP, ViT-MSN, DINOv2) to extract features for both reference and test data.
   Reference features: $\{x_j^r\}_{j=1}^m$
   Test features: $\{x_i^g\}_{i=1}^n$
2: **Compute Nearest Neighbor Distances**:
3: **for** $j = 1$ to $m$ **do**
4:    Compute $\text{NND}_k(x_j^r)$: distance to its $k$-th nearest neighbor in $\{x_h^r\}_{h=1,h\neq j}^m$.
5: **end for**
6: **for** $i = 1$ to $n$ **do**
7:    Compute $\text{NND}_k(x_i^g)$: distance to its $k$-th nearest neighbor in $\{x_h^g\}_{h=1,h\neq i}^n$.
8: **end for**
9: **Compute Per-Point PRDC Metrics for Test Data**:
10: **for** $i = 1$ to $n$ **do**
11:    Compute per-point metrics for $x_i^g$ relative to $\{x_j^r\}_{j=1}^m$:
   1. **Precision per point**:
$$\text{precision}_{pp}^{(i)} = \mathbb{1}\left(x_i^g \in S(\{x_j^r\}_{j=1}^m)\right)$$
   where $S(\{x_j^r\}_{j=1}^m) = \bigcup_{j=1}^m B\left(x_j^r, \text{NND}_k(x_j^r)\right)$.
   2. **Recall per point**:
$$\text{recall}_{pp}^{(i)} = \frac{1}{m}\sum_{j=1}^m \mathbb{1}\left(x_j^r \in B\left(x_i^g, \text{NND}_k(x_i^g)\right)\right)$$
   3. **Density per point**:
$$\text{density}_{pp}^{(i)} = \frac{1}{km}\sum_{j=1}^m \mathbb{1}\left(x_i^g \in B\left(x_j^r, \text{NND}_k(x_j^r)\right)\right)$$
   4. **Coverage per point**:
$$\text{coverage}_{pp}^{(i)} = \mathbb{1}\left(\min_{j=1,\ldots,m}\|x_i^g - x_j^r\| < \text{NND}_k(x_i^g)\right)$$

12:    Assemble feature vector $\phi^{(i)} = \left[\text{precision}_{pp}^{(i)}, \text{recall}_{pp}^{(i)}, \text{density}_{pp}^{(i)}, \text{coverage}_{pp}^{(i)}\right]$
13: **end for**
14: **Prepare Reference Training and Validation Sets**:
15: Split $\{x_j^r\}_{j=1}^m$ into training set $\{x_j^{\text{train}}\}$ and validation set $\{x_j^{\text{valid}}\}$.
16: **Compute Per-Point PRDC Metrics for Reference Training Data**:
17: **for** each $x_j^{\text{train}}$ **do**
18:    Compute per-point metrics $\phi_{\text{ref}}^{(j)}$ following similar steps as above, relative to $\{x_h^{\text{train}}\}_{h=1}^{m_{\text{train}}}$
19: **end for**
20: **Train Anomaly Detection Models**:
21: Use the reference per-point metrics $\{\phi_{\text{ref}}^{(j)}\}$ to train unsupervised anomaly detection models: One-Class SVM (OCSVM), Kernel Density Estimation (KDE), & Gaussian Mixture Model (GMM).
22: **Evaluate Models on Test Data**:
23: **for** $i = 1$ to $n$ **do**
24:    Compute anomaly scores $s^{(i)}$ for $\phi^{(i)}$ using the trained models. Use it to assign Ground Truth Labels, and calculate AUROC & FPR@95
25: **end for**

---

**Notes**:

- $\mathbb{1}(\cdot)$ is the indicator function, returning 1 if the condition is true, and 0 otherwise.
- $B(x, r)$ denotes a ball (in Euclidean space) centered at $x$ with radius $r$.
- $\text{NND}_k(x)$ is the distance from point $x$ to its $k$-th nearest neighbor.
- $S(\{x_j^r\}_{j=1}^m) = \bigcup_{j=1}^m B\left(x_j^r, \text{NND}_k(x_j^r)\right)$ represents the union of balls around each reference point $x_j^r$ with radius $\text{NND}_k(x_j^r)$.

## G  COMPARATIVE EVALUATION

To contextualize the strong state-of-the-art performance achieved by Forte, we present the following infographics showing the evolution of methods and their performance on benchmark tasks.

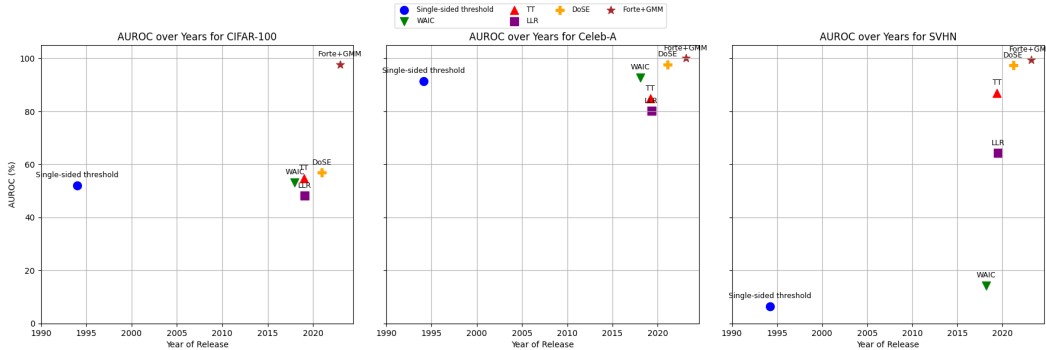

Figure 15: AUROC performance of various unsupervised OOD detection methods over years for the CIFAR-10 (In-distribution) and the CIFAR-100(OOD), Celeb-A(OOD) and SVHN(OOD) dataset. The figure illustrates the progression of anomaly detection techniques, with methods represented using unique markers and colors.

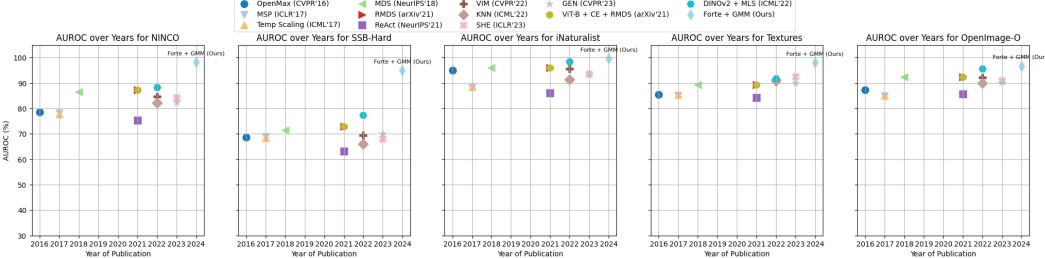

Figure 16: Comparison of AUROC performance across various supervised out-of-distribution detection methods and datasets from the OpenOOD leaderboard. The figure presents results for five datasets (NINCO, SSB-Hard, iNaturalist, Textures, and OpenImage-O) with each subplot showcasing the progression of AUROC scores over the years. Each method is represented with a unique marker and color. Our method "Forte + GMM" is highlighted for its superior performance, demonstrating strong state-of-the-art results across all datasets. The x-axis represents the publication year, while the y-axis denotes the AUROC (%) scores.

## H  MISCELLANEOUS RELATED WORKS

**Data augmentation using generative models,** particularly diffusion models, has shown promise in improving model performance and generalization. Methods include using pretrained models to generate variations of existing data (Luzi et al., 2022; Sariyildiz et al., 2022), fine-tuning models on specific subjects or concepts (Gal et al., 2022; 2023; Kawar et al., 2023), and generating synthetic

datasets for downstream tasks (Shipard et al., 2023; Roy et al., 2022; Packhäuser et al., 2023; Ghalebikesabi et al., 2023; Zhang et al., 2024b; Trabucco et al., 2023; Karras et al., 2020; Bansal & Grover, 2023; Akrout et al., 2023). Other notable examples include TransMix Chen et al. (2022) and MixPro Zhao et al. (2023), which have demonstrated strong performance on the ImageNet classification task, However, the effectiveness of these methods is lower compared to traditional data augmentation and retrieval baselines (Zietlow et al., 2022; Azizi et al., 2023; Burg et al., 2023). Potential biases introduced by synthetic data and the need to detect out-of-distribution generated samples should be considered when employing these techniques.

The concept of score-based likelihood maximization is fundamental to diffusion models, inherently guiding the reverse generation process towards maximizing the likelihood. This maximization process likely pushes the generated images closer to the dominant modes of the distribution Song et al. (2021); Song & Ermon (2020), which has been discussed but has not been explicitly demonstrated as a failure mode by Yamaguchi & Fukuda (2023).

