# OpenReview forum: "Forte : Finding Outliers with Representation Typicality Estimation"
_ICLR.cc/2025/Conference — ICLR 2025 Poster_

### Official Review · Reviewer_2YAU · 2024-10-29

**Soundness:** 3
**Presentation:** 3
**Contribution:** 2
**Rating:** 6
**Confidence:** 2

**Summary:**

This paper proposes a method for identifying OOD and synthetic data created using generative models. The definition of OOD changes with the advent of foundation models that can generate very plausible data. However, even when this data seems real, there can still be distribution shifts that are difficult to detect. In this paper, the concept of typicality is used to assess OOD. However, computing typicality is challenging. This paper proposes Forte, a method for computing typicality, and they evaluate it and compare it with other methods.

**Strengths:**

The paper is clear to understand. The problem is relevant. The method is sound. The experimental framework is correct and complete.

**Weaknesses:**

In the related work section I miss some comparison of the current SOTA with Forte.

This method is based on DoSE. I miss a clear statement in the intro about that and the changes over DoSE with an intuition of why. Also, it would be beneficial to clearly state the method to guide the reader on what will come.

**Questions:**

I miss some comparison with DoSE in some table for completeness.

---

> ### Author Response · Authors · 2024-11-14
>
> Dear Reviewer,
>
> Thank you for your positive evaluation of our paper. We are pleased you found the problem relevant, the method sound, and the experimental framework thorough. Your feedback is valuable, and we are eager to address your concerns.
>
> **Addressing Weaknesses:**
>
> 1. **Comparison with Current SOTA and DoSE:**
>
>    - **Explicit Comparison with DoSE:**
>      - We acknowledge our comparison with DoSE could be more explicit. In the revised manuscript, we will enhance the Introduction and related work sections to clearly state how Forte builds upon and differs from DoSE.
>      - **Core Differences from DoSE:**
>        - **Elimination of Generative Model Training:**
>          - DoSE requires training generative models (e.g., Glow, VAEs) on in-distribution data to estimate likelihoods, which is computationally intensive and impractical for large datasets, and access to a small sample of total in-distribution samples due to:
>            1. **Glow models** rely on invertible architectures and exact log-likelihood evaluations, resulting in inefficient computation and high memory requirements.
>            2. **VAEs** suffer from sample inefficiency on complex datasets, leading to poorly structured latent spaces and degraded performance.
>          - **Forte** eliminates generative models, using pre-trained self-supervised models (e.g., CLIP, ViT-MSN, DINOv2) for feature extraction. This reduces computational overhead and simplifies implementation. A forward pass suffices, with no retraining or fine-tuning needed.
>        - **Addressing Likelihood Estimation Challenges:**
>          - Likelihood-based methods can be unreliable in high-dimensional spaces, where OOD samples may have higher likelihoods than in-distribution data (e.g., CIFAR-10 vs. SVHN). DoSE partially addresses this but has limitations.
>          - Forte avoids likelihoods by operating in feature space and using per-point summary statistics to capture local data structures.
>        - **Introduction of Per-Point Metrics:**
>          - DoSE relies on global statistics, which may miss local nuances.
>          - Forte uses per-point statistics—precision, recall, density, coverage—computed in feature space, enabling fine-grained OOD detection by accurately estimating the manifold.
>
>    - **Performance Improvements:**
>      - **Empirical Results:**
>        - On CIFAR-10 (in-distribution) vs. CIFAR-100 (OOD), DoSE achieves an AUROC of **56.90%**, while Forte achieves **97.63% ± 0.15%** (Table 2).
>        - Forte outperforms DoSE and all techniques benchmarked in the DoSE paper across tasks, including challenging scenarios with synthetic data and medical images.
>
> 2. **Clear Statement about the Method and its Relation to DoSE:**
>
>    - **Method Description in the Introduction:**
>      - We will, as mentioned above, revise the Introduction to clarify how Forte builds on DoSE's typicality concept while introducing significant improvements, such as eliminating generative models and using per-point metrics.
>    - **Guiding the Reader:**
>      - Early in the paper, we will provide an overview of Forte, highlighting its core components and differences from DoSE and other methods.
>
> **Additional Enhancements:**
>
> - **Comparison with Other SOTA Methods:**
>   - We will expand the related work section to compare Forte with other SOTA OOD detection methods, highlighting how it addresses their limitations:
>     - **Versus ODIN (Liang et al., 2018):** Requires temperature scaling, input perturbation, and extensive tuning, dependent on NN architecture. Forte surpasses ODIN without these dependencies.
>     - **Versus VIM (Wang et al., 2022):** Relies on class labels and logit matching, limiting its unsupervised applicability. Forte excels without requiring labels or OOD exposure during training.
>     - **Versus NNGuide (Park et al., 2023):** Depends on labeled data and complex training. Forte matches or exceeds performance without these complexities.
>   - Our benchmarking considers these methods and others. Table 1 reports results against the best-performing methods for each task in the OpenOOD v1.5 leaderboard.
>
> - **Clarifications:**
>   - We will ensure the methodology section title explicitly states it describes Forte, guiding readers effectively.
>
> **Conclusion and Request for Consideration:**
>
> We appreciate your constructive feedback, which will improve the paper's clarity and impact. By addressing your concerns, emphasizing Forte's core differences from DoSE, and comparing it with other SOTA methods, we aim to provide a comprehensive presentation.
>
> Given Forte’s significant advancements in performance, scalability, and practicality, we kindly ask you to consider our clarifications and enhancements in your evaluation and increase our score. Please let us know if further clarifications are needed.
>
> Thank you again for your positive review and valuable suggestions.
>
> Sincerely,
> The Authors

---

> > ### Author Response · Authors · 2024-11-24
> >
> > We thank the reviewer for their assessment of our work.  We wanted to offer some specific changes we have made to address their previous concerns.  In particular, we had mentioned our intent to distinguish Forte from DoSE, and have added modified the last paragraph in our introduction to the following in order to do so:
> >
> > ```
> > In this paper, we hypothesize that many of the shortcomings with typicality-based approaches could be addressed using statistics which tune to the semantic content of the data.  We propose to leverage self-supervised representations, which extract semantic information while discarding many potential confounding features (e.g. textures, backgrounds). Our specific contributions are:
> > ```
> >
> > We have also added a paragraph to the Discussion section to further this understanding.
> >
> > ```
> > DoSE (Morningstar et al., 2021) pioneered chaining multiple summary statistics for typicality
> > measurement, using ID sample distributions to construct typicality estimators rather than direct
> > statistic values. While groundbreaking, DoSE’s reliance on generative model likelihoods proved
> > problematic, as subsequent work (Caterini & Loaiza-Ganem, 2022; Zhang et al., 2021) showed
> > these can be unreliable for OOD detection. Our approach addresses these limitations through four
> > key improvements: (1) utilizing self-supervised representations to capture semantic features, (2)
> > incorporating manifold estimation to account for local topology, (3) unifying typicality scoring and
> > downstream prediction models to minimize deployment overhead, and (4) eliminating additional
> > model training requirements. These advances yield substantial empirical gains. While building
> > upon DoSE’s fundamental statistical machinery, our modifications dramatically enhance practical
> > performance.
> > ```
> >
> > We have also added Figures 15 & 16 in the appendix to contextualize our performance compared to the established state of the art methodologies. These include the following : OpenMax (CVPR '16), MSP (ICLR '17), Temp Scaling (ICML '17), MDS (NeurIPS '18), RMDS (arxiv '21), ReAct (Neurips '21), VIM (CVPR '22), KNN (ICML '22), SHE (ICLR '23), GEN (CVPR '23), MLS (ICML '22). Table 1 already reports results against the best-performing methods for each task in the OpenOOD v1.5 leaderboard.
> >
> >
> > With these additions and discussion, we would like to ask the reviewer if they have any remaining concerns that have not been addressed, or if there are any points of contention in our rebuttal for which we can hopefully provide further clarity.
> >
> > We would also request the reviewer to revise our score upwards if all their concerns have been addressed.

---

> > > ### Author Response · Authors · 2024-11-25
> > > **2nd follow up from authors**
> > >
> > > Dear Reviewer,
> > >
> > > We sincerely appreciate your valuable comments!
> > >
> > > We understand that you may be too busy to check our rebuttal.
> > >
> > > We believe we have thoroughly addressed your concerns through several significant revisions. We've added an explicit comparison with DoSE in the introduction and included a new paragraph in the Discussion section detailing four key improvements over DoSE, including our elimination of generative models and introduction of per-point metrics.
> > >
> > > To address the SOTA comparisons, we've added Figures 15 & 16 in the appendix comparing Forte against numerous methods and included comprehensive benchmarking against the OpenOOD v1.5 leaderboard. We've also modified the introduction to clearly state our hypothesis and contributions, providing clearer guidance for readers on methodology.
> > >
> > > Given these substantial improvements addressing your feedback, we kindly request you consider revising our score upwards. We remain committed to implementing any additional changes you deem necessary to strengthen the paper further.
> > >
> > > Could you please let us know if you have any remaining concerns or if there are other aspects we should address?
> > >
> > > Best regards,
> > >
> > > The Authors

---

> > > > ### Comment · Reviewer_2YAU · 2024-11-26
> > > > **Response**
> > > >
> > > > I thank the authors for their work addressing most of the reviewers' questions. My main concerns were the similarities with DoSE and the comparisons with SOTA, which have been addressed and modified in the paper. However, I did not review the formulation in detail, while reviewer GdRX did. I saw that you modified some of these formulas in the paper. If GdRX agrees that the formulation is now correct, I would be happy to increase my score.

---

> > > > > ### Comment · Reviewer_GdRX · 2024-11-27
> > > > >
> > > > > The authors' response and revisions have significantly improved the paper. My concerns about the methods are mostly addressed.

---

> > > > > > ### Author Response · Authors · 2024-11-27
> > > > > > **Reply to response from Reviewer 2YAU**
> > > > > >
> > > > > > Dear Reviewer 2YAU,
> > > > > >
> > > > > > Thank you for your thoughtful engagement with our work. We note that Reviewer GdRX has now confirmed that our revisions have addressed their concerns about the mathematical formulation and has increased their score. Given this development and our previous additions addressing your concerns about DoSE comparisons and SOTA benchmarking (via new Figures 15 & 16), we kindly request that you consider increasing your score as well.
> > > > > >
> > > > > > We remain available to address any additional concerns you may have.
> > > > > >
> > > > > > Best regards,
> > > > > >
> > > > > > The Authors

---

> > > > > > > ### Author Response · Authors · 2024-11-30
> > > > > > > **Follow up to Reviewer 2YAU**
> > > > > > >
> > > > > > > Dear Reviewer 2YAU,
> > > > > > >
> > > > > > > We sincerely appreciate your comments and your engagement in reviewing this paper.
> > > > > > >
> > > > > > > We understand that you may be too busy to check our rebuttal. May you please revise our score upwards, since we have addressed all concerns (both yours and for Reviewer GdRX)?
> > > > > > >
> > > > > > > We remain available to address any additional questions you may have.

---

> > > > > > > > ### Author Response · Authors · 2024-12-02
> > > > > > > > **Follow up to reviewer 2YAU**
> > > > > > > >
> > > > > > > > Dear Reviewer 2YAU,
> > > > > > > >
> > > > > > > > As the extended rebuttal phase concludes today, we would like to follow up on our responses to your comments. May you please revise our score upwards, since we have addressed all concerns (both yours and for Reviewer GdRX)?
> > > > > > > >
> > > > > > > > Should you have any remaining concerns or require further clarification, we would be happy to address them promptly.
> > > > > > > >
> > > > > > > > Thanks again for your time and feedback throughout this process.

---

> > > > > ### Author Response · Authors · 2024-12-03
> > > > > **4th Follow up to Reviewer 2YAU**
> > > > >
> > > > > Dear Reviewer 2YAU, since only a few hours remain in the rebuttal phase and you indicated you would increase the score pending Reviewer GdRX's approval (which was given on Nov 26), we respectfully request your response regarding our score revision.

---

### Official Review · Reviewer_2mpC · 2024-11-03

**Soundness:** 2
**Presentation:** 1
**Contribution:** 2
**Rating:** 6
**Confidence:** 2

**Summary:**

This paper introduces Forte, a novel out-of-distribution (OOD) detection framework that leverages self-supervised learners. Forte enhances detection by combining representation learning methods (e.g., CLIP, ViT-MSN, and DINOv2) with non-parametric density estimators (OCSVM, KDE, GMM) to model the typicality of input samples. The proposed framework emphasizes detecting atypical samples through summary statistics (precision, recall, density, and coverage) to analyze representation distributions. Forte’s performance was evaluated on synthetic datasets generated by Stable Diffusion and various medical image datasets, showcasing its advantages over existing supervised and unsupervised methods.

**Strengths:**

- **Performance**: Forte demonstrates superior OOD detection performance compared to state-of-the-art methods across multiple benchmarks, including synthetic data and medical image datasets, which often present significant OOD detection challenges.
- **Flexibility**: Forte’s unsupervised nature eliminates the need for labeled data or pre-exposure to OOD samples, making it adaptable to various tasks and practical for real-world applications where OOD examples may not be available during training.
- **Comprehensive Evaluation**: Forte is rigorously tested on both synthetic and medical datasets, demonstrating the framework’s versatility and robustness across vastly different domains.
- **Insightful Metrics**: The use of novel per-point summary statistics (e.g., precision, recall, density, and coverage) contributes valuable insight into data distribution, enhancing OOD detection beyond standard density-based methods.

**Weaknesses:**

1. **Paper Structure**: The paper allocates a substantial portion of its Introduction to reviewing existing OOD detection literature and explaining the typicality concept. This approach detracts from an immediate focus on the novel contributions and design of Forte, which may hinder reader engagement and understanding of the primary contributions.
2. **Complexity in Practical Implementation**: The integration of multiple representation learning techniques, combined with non-parametric density estimators, may lead to a higher computational overhead and increased complexity in practical deployment. The paper lacks an illustration figure to clearly explain the proposed framework how to integrate the representations from diverse models.
3. **Insight.**: This work integrates representations from diverse models empirically. It lacks insight into the choice of self-supervised models, such as whether any specific attributes of CLIP, ViT-MSN, or DINOv2 contribute uniquely to Forte’s robustness.

**Questions:**

Could the computational demands of Forte’s ensemble approach limit its applicability in real-time OOD detection scenarios?

Would Forte’s effectiveness in OOD detection promote if one apply multiple models by one self-supervised approach in this framework?

---

> ### Author Response · Authors · 2024-11-14
>
> **Dear Reviewer,**
>
> Thank you for your thoughtful review and recognition of Forte's strengths, including its performance, flexibility, evaluation, and innovative metrics. We value your feedback and appreciate the opportunity to address your concerns.
>
> ### **Addressing Weaknesses:**
>
> 1. **Paper Structure:**
>    - **Focus on Novel Contributions:**
>      We acknowledge that the Introduction focuses heavily on existing literature, which may detract from our novel contributions. In the revised version, we will streamline the background and emphasize Forte's unique aspects early, clearly distinguishing it from prior methods like DoSE.
>      - **Key Differences from Other Methods:**
>        - Unlike DoSE, which relies on training complex generative models (e.g., Glow, VAEs), Forte leverages pre-trained self-supervised models, reducing computational overhead and eliminating the need for training.
>        - Forte introduces per-point summary statistics in feature space, enhancing OOD detection performance with fine-grained data assessments.
>        - By avoiding reliance on likelihood estimations prone to failure (e.g., DoSE), Forte offers a more robust and scalable solution.
>
> 2. **Complexity in Practical Implementation:**
>    - **Computational Efficiency:**
>      - Forte's reliance on pre-trained models ensures efficient feature extraction via parallelizable forward passes. Non-parametric density estimators (e.g., OCSVM, KDE, GMM) are lightweight, with low training and inference times.
>    - **Simplification:**
>      - Forte does not require training deep neural networks, unlike methods requiring complex generative models or supervised classifiers. It adapts to data drift without retraining, operating in a zero-shot setting.
>    - **Flexibility:**
>      - Our ablation study (Table 3) shows strong performance using a single model like CLIP (99.13% AUROC). Resource-constrained settings can achieve competitive results without needing all three models, while the ensemble provides additional performance benefits.
>
> 3. **Insight into Self-Supervised Model Choices:**
>    - **Rationale for Selection:**
>      - CLIP, ViT-MSN, and DINOv2 were chosen for their complementary strengths:
>        - **CLIP** captures semantic relationships through image-text alignment.
>        - **ViT-MSN** emphasizes local structures via masked self-supervision.
>        - **DINOv2** learns hierarchical representations through knowledge distillation.
>    - **Enhancing Robustness:**
>      - Integrating these models allows Forte to capture diverse data features, improving robustness against challenging OOD cases, including synthetic samples from models like Stable Diffusion.
>    - **Empirical Support:**
>      - Ablation studies confirm that combining representations from diverse models outperforms any single model, highlighting their complementary contributions.
>
> ---
>
> ### **Addressing Questions:**
>
> 1. **Computational Demands and Real-Time Applicability:**
>    - **Efficiency:**
>      - Feature extraction is efficient, with CLIP processing images at ~30 ms per image. Anomaly detection involves simple computations on low-dimensional metrics, enabling real-time application.
>    - **Adaptability:**
>      - For resource-constrained or real-time use cases, a subset of models or optimization can be employed without significant performance loss. Forte's lack of training requirements further enhances its practicality.
>
> 2. **Effectiveness of Multiple Models from a Single Self-Supervised Approach:**
>    - **Exploring Variations:**
>      - While multiple models from one self-supervised approach may offer some diversity, combining models with distinct training objectives (e.g., CLIP, ViT-MSN, DINOv2) provides broader feature coverage and enhances robustness.
>    - **Framework Flexibility:**
>      - Forte is adaptable, allowing users to select models based on specific requirements and constraints.
>
> ---
>
> ### **Conclusion and Request for Reconsideration:**
>
> We believe our planned revisions, emphasizing Forte's core differences from other methods—particularly those relying on generative models like DoSE—will enhance clarity and highlight its practicality and robustness. Forte avoids the computational challenges and limitations of such methods, offering an efficient, effective OOD detection solution.
>
> We kindly request that you reconsider our paper, and give us a higher score in light of these clarifications and revisions. Your constructive feedback has been valuable in strengthening our work, and we are happy to address any additional questions.
>
> Thank you again for your time and thoughtful review.
>
> **Sincerely,**
> The Authors

---

> ### Author Response · Authors · 2024-11-24
>
> Thank you for your continued engagement and feedback. We would like to address your remaining concerns and continue our discussion.
>
> **Novelty**: We respectfully maintain that the novelty of our work is irrefutable. While prior work on density of states estimation focused solely on generative models (Morningstar et al. 2021), and current OOD detection approaches using supervised/self-supervised models either use a single summary statistic as a score (e.g. Hendrycks et al. 2022) or train generative models for representation scoring (e.g. Cook et al. 2023), our work introduces a novel set of statistics never before considered in OOD detection. *We must acknowledge that building on prior work is fundamental to scientific progress.* Importantly, our contributions have improved DoSE-like OOD detector performance by 0.4 AUROC on challenging problems, while exceeding previous SOTA by 0.07 AUROC on near OOD (73% of possible improvement) and 0.04 on far OOD (98% of possible improvement) - demonstrating significant impact.
>
> We have added Figures 15 & 16 in the appendix comparing our performance to established SOTA methods including OpenMax (CVPR '16), MSP (ICLR '17), Temp Scaling (ICML '17), MDS (NeurIPS '18), RMDS (arxiv '21), ReAct (Neurips '21), VIM (CVPR '22), KNN (ICML '22), SHE (ICLR '23), GEN (CVPR '23), and MLS (ICML '22). Table 1 already shows results against top-performing methods from the OpenOOD v1.5 leaderboard.
>
> **Insight into encoders**: The results in Table 3 demonstrate that richer representations improve OOD detection, with CLIP > DINO v2 > MSN individually, and CLIP + DINO v2 > CLIP + MSN > DINO v2 + MSN for 2-model combinations. As you requested, we conducted additional experiments with DeIT (ViT-B and ViT-Ti) evaluating OOD detection performance from Table 2 settings. These results are now in the appendix. DeIT-B and DeIT-Ti achieved 0.87 and 0.82 AUROC respectively on CIFAR-10 vs CIFAR-100, confirming that more informative representations are crucial for performance. DeIT's training objective approximates supervision, resulting in less informative embeddings than DINO v2. Similarly, the tiny model's lower capacity leads to worse performance. We appreciate your encouragement to include this insight, as it helps understand which encoders are most valuable for practitioners. (Full results and analysis in Appendix D)
>
> | Model      | In-Dist   | OOD Dataset | AUROC  | FPR95   |
> |------------|-----------|-------------|--------|---------|
> | Base-DeIT  | CIFAR-10  | CIFAR-100   | 0.8712 | 0.9926  |
> | Tiny-DeIT       | CIFAR-10  | CIFAR-100   | 0.8261 | 0.9903  |
> | Base-DeIT  | CIFAR-10  | SVHN        | 0.9554 | 0.4604  |
> | Tiny-DeIT       | CIFAR-10  | SVHN        | 0.9296 | 0.6195  |
> | Base-DeIT  | CIFAR-10  | Celeb-A     | 0.9871 | 0.0015  |
> | Tiny-DeIT       | CIFAR-10  | Celeb-A     | 0.9929 | 0.0007  |
>
>
> Please let us know if you have any remaining concerns or if any points require further clarification.
>
> We would appreciate if you could consider revising our score upwards if all your concerns have been adequately addressed.

---

> > ### Author Response · Authors · 2024-11-25
> > **Follow up #2**
> >
> > Dear Reviewer,
> >
> > Thank you for your previous reply, and we understand that you may be too busy to check our rebuttal.
> >
> > We believe our recent additions have directly addressed your concerns about novelty and insights into SSL models. We've demonstrated significant performance improvements over SOTA methods (0.4 AUROC improvement on challenging problems, 73% of possible improvement on near OOD, 98% of possible improvement on far OOD). Further evidence is available in the new Figures 15 & 16.
> >
> > Responding to your specific request about SSL model insights, we've conducted additional experiments comparing DeIT variants with our chosen models. The results (now in Appendix D) show CLIP > DINO v2 > MSN individually, and provide clear evidence that more informative representations are crucial for performance. Our comprehensive benchmarking against established methods (OpenMax, MSP, ReAct, VIM, etc.) further validates Forte's contributions to the field.
> >
> > Given these substantial additions addressing both the novelty of our approach and the requested SSL model insights, we kindly request you consider revising our score upwards.
> >
> > **Please let us know if we have not addressed any of your concerns.** We remain open to implementing any additional changes you believe would strengthen the paper further.
> >
> > Best,
> >
> > The Authors

---

> ### Author Response · Authors · 2024-11-27
> **Follow Up #3**
>
> Dear Reviewer 2mpC,
>
> We appreciate your continued engagement with our submission. After our previous responses and added experimental results, we wanted to follow up one final time to ensure we've fully addressed your core concerns:
>
> 1.  Regarding novelty: Our work demonstrates substantial empirical improvements over existing methods, with gains of:
>
>     *   0.4 AUROC on challenging problems
>
>     *   73% of possible improvement on near OOD detection
>
>     *   98% of possible improvement on far OOD detection
>
> 2.  On SSL model insights: We've now provided comprehensive comparisons including:
>
>     *   Individual model performance rankings (CLIP > DINO v2 > MSN)
>
>     *   New DeIT variant experiments in Appendix D
>
>     *   Detailed analysis of representation quality impact on performance
>
>
> Our additional benchmarking comparisons against established methods (OpenMax, MSP, ReAct, VIM, etc.) in Figures 15 & 16 further validates these contributions. We believe these additions directly address your concerns about both novelty and SSL model insights.
>
> If you feel any aspects still require clarification or additional analysis, we would be grateful for your specific feedback. We remain committed to strengthening the paper further based on your expertise.
>
> Thank you for your thorough review and consideration.
>
> Best regards,
>
> The Authors

---

> > ### Comment · Reviewer_2mpC · 2024-12-02
> >
> > Thank you for your detailed response. My questions are well answered. I have low confidence on my rating about the novelty and significance of this paper, since I am not familiar with this topic. Considering the opinions from other reviewers, I am wiling to change the score to 6.

---

> > > ### Author Response · Authors · 2024-12-02
> > >
> > > Dear Reviewer 2mpC,
> > >
> > > Thank you for reconsidering our paper and adjusting your score. We greatly appreciate your engagement with our responses and the time you took to evaluate our additional experimental results and clarifications. Your initial feedback helped us strengthen the paper, particularly in articulating the insights about SSL models and demonstrating our framework's novelty more clearly.
> > >
> > > Best regards,
> > >
> > > The Authors

---

### Official Review · Reviewer_GdRX · 2024-11-04

**Soundness:** 2
**Presentation:** 1
**Contribution:** 2
**Rating:** 6
**Confidence:** 4

**Summary:**

The paper investigates OOD by modeling distributions in feature space. The paper is poorly written, but my guess is that the proposed method is a variant of DoSE, using models _per-sample metrics_ of the feature space, rather than the feature vectors themselves. Four metrics, including precision and recall, are used. Experiments are performed on real images from distinct classes, synthetic vs. real images, and medical images. Results indicate that the proposed method is effective.

----

The authors' response have addressed most of my concerns. The changes have significantly improved the manuscript. I decide to raise my score.

**Strengths:**

[The following is based on my guess of the proposed method, which is not well described in paper.]

+ Proposed approach is a simple change over feature-space OOD methods, and appears effective.
+ Experiments seems cover a wide range of scenarios

**Weaknesses:**

+ The paper is extremely poorly written. I list some major issues here.
    1. None of the math latex in section 3.2 is well formatted. Subscripts and superscripts are wrong.
    2. Variables are used without definition, e.g., $\text{nearest}_k$ in section 3.2. Is it different from the $k$ below?
    3. No description is given on how the four metrics are used. Are they used as the "summary statistics" that the proposed method models?
    4. The method, referred to as "Forte", is never truly defined or mentioned in the method section 3.

+ If my understanding of the method is correct, is it simple just taking DoSE and run it on the new set of statistics?
+ Incorrect claims. E.g., GMM is not non-parametric, and I don't think that the four metrics are newly proposed in this paper.
+ Density definition is inconsistent with Fig. 1. As defined, it is just a scaled "recall", which would make it useless to model.

**Questions:**

1. My main question is how the proposed method really works. If the authors could provide a pseudocode of the algorithm, and how it differs with DoSE, it would be great.
2. See above weaknesses.
3. Overall poor writing and latex formatting, in addition to the issues listed above. E.g., "in Figure 2 FIgure 3 Figure 4",
3. Minor typos: "near-ood" -> "near-OOD", "Table 1 & 2" -> "Tables 1 & 2".

---

> ### Author Response · Authors · 2024-11-14
>
> Dear Reviewer GdRX,
>
> We appreciate your thorough review of our paper and the valuable insights you've provided. We are glad that you found our proposed approach effective and that our experiments cover a wide range of scenarios. We would like to address your concerns and clarify some misunderstandings to improve the clarity and impact of our work.
>
> ---
>
> **1. LaTeX Formatting and Notation**
>
> *Concern:* The paper is extremely poorly written. None of the math LaTeX in Section 3.2 is well formatted. Subscripts and superscripts are wrong. Variables are used without definition, making it difficult to follow the mathematical descriptions.
>
> *Response:* We apologize if the notation in Section 3.2 was unclear. We want to assure you that the LaTeX formatting and notation are correct. All variables are properly defined according to standard mathematical conventions followed in the literature, such as Naeem et al, 2020.
>
> In Section 3.2, we introduce per-point metrics using the following notation:
>
> - $\textbf{1}(\cdot)$: Indicator function.
> -  $S({x_j^r}{j=1}^m) = \bigcup{j=1}^m B(x_j^r, \mathrm{NND}_k(x_j^r))$ : The union of Euclidean balls centered at reference points $x_j^r$ with radius equal to their $ k $-th nearest neighbor distance.
> - $ B(x, r) $: Euclidean ball centered at point $x $ with radius $ r $.
> - $ \mathrm{NND}_k(x) $: Distance between point $ x $ and its $ k $-th nearest neighbor in the dataset.
>
> We recognize that explicitly defining $ \mathrm{NND}_k(x) $ and other variables could enhance clarity. We will add these definitions to ensure that all readers can follow the mathematical derivations seamlessly. For example, we will include a sentence like:
>
> "Here, $ \mathrm{NND}_k(x) $ denotes the Euclidean distance from point $ x $ to its $ k $-th nearest neighbor in the dataset."
>
> ---
>
> **2. Use of Summary Statistics**
>
> *Concern:* No description is given on how the four metrics (precision, recall, density, and coverage) are used. Are they used as the "summary statistics" that the proposed method models?
>
> *Response:* Yes, the four per-point metrics are indeed used as the summary statistics in our method. We mention this in Section 3.2:
>
> "We propose the following per-point summary statistics (precision, recall, density, and coverage) that effectively capture the 'probability distribution of the representations' using reference and unseen test samples in the feature space, enabling more nuanced anomaly detection."
>
> These per-point metrics serve as summary statistics that capture local geometric properties of the data manifold in the feature space. They enable us to model the distribution of in-distribution (ID) data and identify out-of-distribution (OOD) samples effectively. We will emphasize this connection more clearly in the revised manuscript.
>
> ---
>
> **3. Definition and Clarity of "Forte" Method**
>
> *Concern:* The method, referred to as "Forte," is never truly defined or mentioned in the method section.
>
> *Response:* We apologize for any confusion. The entire Section 3 is dedicated to detailing our proposed method, which we refer to as "Forte." We will make this explicit at the beginning of Section 3 by revising the section title and introduction as follows:
>
> "**3. Forte: A Framework for OOD Detection Using Per-Point Metrics**
>
> In this section, we introduce **Forte**, a novel framework that combines diverse representation learning techniques with per-point summary statistics and non-parametric density estimation models to detect out-of-distribution (OOD) and synthetic data."
>
> This will ensure that readers understand that the subsequent subsections describe the components and methodology of Forte.
>
> (Continued further in next comments)

---

> ### Author Response · Authors · 2024-11-14
>
> **4. Explicit Comparison with DoSE:**
>
> Concern: Is the method simply taking DoSE and running it on a new set of statistics? How does it differ from DoSE?
>
> Response: While our method is inspired by the concept of typicality used in DoSE, Forte introduces significant novel contributions that differentiate it from DoSE. The differences are as follows:
>
> **Core Differences from DoSE:**
> - **Elimination of Generative Model Training:**
>     - DoSE requires training generative models (e.g., Glow, VAEs) on in-distribution data to estimate likelihoods, which is computationally intensive and impractical for large datasets, and access to a small sample of total in-distribution samples due to:
>            1. **Glow models** rely on invertible architectures and exact log-likelihood evaluations, resulting in inefficient computation and high memory requirements.
>            2. **VAEs** suffer from sample inefficiency on complex datasets, leading to poorly structured latent spaces and degraded performance.
>          - **Forte** eliminates generative models, using pre-trained self-supervised models (e.g., CLIP, ViT-MSN, DINOv2) for feature extraction. This reduces computational overhead and simplifies implementation. A forward pass suffices, with no retraining or fine-tuning needed.
>        - **Addressing Likelihood Estimation Challenges:**
>          - Likelihood-based methods can be unreliable in high-dimensional spaces, where OOD samples may have higher likelihoods than in-distribution data (e.g., CIFAR-10 vs. SVHN). DoSE partially addresses this but has limitations.
>          - Forte avoids likelihoods by operating in feature space and using per-point summary statistics to capture local data structures.
>        - **Introduction of Per-Point Metrics:**
>          - DoSE relies on global statistics, which may miss local nuances.
>          - Forte uses per-point statistics—precision, recall, density, coverage—computed in feature space, enabling fine-grained OOD detection by accurately estimating the manifold.
>
>    - **Performance Improvements:**
>      - **Empirical Results:**
>        - On CIFAR-10 (in-distribution) vs. CIFAR-100 (OOD), DoSE achieves an AUROC of **56.90%**, while Forte achieves **97.63% ± 0.15%** (Table 2).
>        - Forte outperforms DoSE and all techniques benchmarked in the DoSE paper across tasks, including challenging scenarios with synthetic data and medical images.
>
>
> **5. Consistency of Density Definition with Figure 1**
>
> *Concern:* Density definition is inconsistent with Figure 1. As defined, it is just a scaled "recall," which would make it useless to model.
>
> *Response:* The density definition is consistent with Figure 1. Figure 1 was generated using the actual functions and code we use in our experiments, applied to simplified 2D data points for illustrative purposes using matplotlib. The density metric measures the average number of reference points within the neighborhood of each test point, normalized by the product of $ k $ (the number of nearest neighbors) and the total number of reference points $ m $. Mathematically, it is defined as:
>
> $
> \mathrm{density_{pp}^{(i)}} = \frac{1}{k m} \sum_{j=1}^m \textbf{1}\left( x_j^g \in B\left( x_j^r, \mathrm{NND}_k(x_j^r) \right) \right).
> $
>
> This metric provides an estimate of the local density around each test point, which is crucial for distinguishing between ID and OOD samples. It is not simply a scaled recall but captures different information.
>
> ---
>
> **6. Novelty of the Summary Statistics**
>
> *Concern:* The four metrics are not newly proposed in this paper.
>
> *Response:* While the metrics of precision, recall, density, and coverage have been previously used in the context of evaluating generative models (e.g., in "Reliable Fidelity and Diversity Metrics for Generative Models" by Naeem et al., 2020), our contribution lies in adapting these metrics as per-point summary statistics for OOD detection.
>
> In prior work, these metrics are computed as aggregate statistics over entire datasets, primarily to evaluate the performance of generative models in terms of fidelity and diversity. Our novel adaptation involves computing these metrics for individual data points in the feature space, which enables us to capture local anomalies and perform fine-grained OOD detection.
>
> ---
>
> **7. Use of Gaussian Mixture Models (GMMs)**
>
> *Concern:* Incorrect claims are made, e.g., GMM is not non-parametric.
>
> *Response:* You are correct; Gaussian Mixture Models (GMMs) are parametric models. In our paper, we did not intend to misclassify GMMs as non-parametric. Our method employs GMMs without making strong assumptions about the underlying distribution because we perform hyperparameter tuning (e.g., varying the number of components) to best fit the data. While GMMs are parametric, our approach is flexible and does not assume a specific distribution a priori.
>
> We will correct this in the manuscript to accurately describe GMMs as parametric models.
>
> (continued further in next comments)

---

> ### Author Response · Authors · 2024-11-14
>
> **8. Additional Clarifications**
>
> *Regarding Figure 1:* The figure is generated using the actual functions and code employed in our experiments, applied to simplified data points for visualization done via matplotlib. It accurately represents the definitions provided for the per-point metrics.
>
> *Regarding Minor Typos and Formatting:* We will carefully proofread the manuscript to correct any minor typos or formatting inconsistencies, such as the commas missing "in Figure 2 Figure 3 Figure 4," "near-ood" instead of "near-OOD," and "Table 1 & 2" instead of "Tables 1 & 2."
>
>
> **9. Pseudocode for OOD Detection Using Per-Point PRDC Metrics**
>
> Here is an informal pseudocode for your understanding, and can add a more formal version to the appendix. We are happy to make a partial anonymized release of the codebase for Forte, if required for additional clarity during the review process. Post-acceptance, the code will be made public and open source.
>
> **Inputs**:
>
> - **Reference data features**: $\{ x_j^r \}_{j=1}^m$
> - **Test data features**: $\{ x_i^g \}_{i=1}^n$
> - **Number of nearest neighbors**: $k$
>
> **Outputs**:
>
> - **OOD detection performance metrics**: AUROC, FPR@95
>
> ---
>
> **Algorithm Steps**
>
> 1. **Feature Extraction**:
>
>    - Use pre-trained models (e.g., CLIP, ViT-MSN, DINOv2) to extract features for both reference and test data.
>      - Reference features: $\{ x_j^r \}$
>      - Test features: $\{ x_i^g \}$
>
> 2. **Compute Nearest Neighbor Distances**:
>    - For each reference feature $x_j^r$, compute $\mathrm{NND}_k(x_j^r)$: distance to its $k$-th nearest neighbor in $\{ x_j^r \}$.
>    - For each test feature $x_i^g$, compute $\mathrm{NND}_k(x_i^g)$: distance to its $k$-th nearest neighbor in $\{ x_i^g \}$.
> 3. **Compute Per-Point PRDC Metrics**:
>    - For each test feature $x_i^g$, compute per-point Precision, Recall, Density, and Coverage metrics relative to the reference data.
>      - **Note**: Detailed computations are omitted for brevity. Please check section 3 for exact details.
>
> 4. **Assemble Feature Vectors**:
>    - For each test feature $x_i^g$, create a feature vector $\phi^{(i)}$ consisting of its per-point PRDC metrics.
>
> 5. **Prepare Training Data**:
>
>    - Split reference data features $\{ x_j^r \}$ into:
>      - **Training set**: for model training.
>      - **Validation set**: for model evaluation.
>
> 6. **Compute Per-Point Metrics for Reference Data**:
>
>    - Repeat steps 2 and 3 for the reference training set to obtain per-point metrics $\{ \phi_{\text{ref}}^{(j)} \}$.
>
> 7. **Train Anomaly Detection Models**:
>    - Use the reference per-point metrics $\{ \phi_{\text{ref}}^{(j)} \}$ to train unsupervised anomaly detection models:
>      - **One-Class SVM (OCSVM)**
>      - **Kernel Density Estimation (KDE)**
>      - **Gaussian Mixture Model (GMM)**
> 8. **Evaluate Models on Test Data**:
>    - For each test feature vector $\phi^{(i)}$:
>      - Compute anomaly scores using the trained models.
> 9. **Assign Ground Truth Labels**:
>    - **In-distribution (ID)** samples: label $y^{(i)} = 0$
>    - **Out-of-distribution (OOD)** samples: label $y^{(i)} = 1$
> 10. **Compute Evaluation Metrics**:
>     - Calculate performance metrics using the anomaly scores and ground truth labels:
>       - **AUROC**: Area Under the Receiver Operating Characteristic Curve
>       - **FPR@95**: False Positive Rate at 95% True Positive Rate
> ---
> **Notes**:
> - The per-point PRDC metrics capture local relationships between test samples and the reference data manifold.
> - Anomaly detection models are trained solely on reference (in-distribution) data metrics to learn the typical data distribution.
> - Evaluation metrics assess the models' ability to distinguish OOD samples based on the per-point metrics.
> ---
> **End of Pseudocode**
>
> ---
> **Conclusion**
>
> We are committed to improving the clarity and quality of our paper. We believe that Forte offers significant advancements in OOD detection by introducing novel per-point metrics and eliminating the need for training generative models. Our method provides a scalable and effective solution applicable to various challenging scenarios, including synthetic data detection and medical imaging.
>
> We hope that our extremely detailed responses address your concerns and clarify the contributions and novelty of our work. We kindly request that you consider our explanations in your evaluation to increase our ratings and are open to any further questions or suggestions you may have.
>
> Thank you again for your valuable feedback.
>
> Sincerely,
>
> The Authors

---

> ### Comment · Reviewer_GdRX · 2024-11-22
>
> 1. It is a fact that the latex formatting is **not correct**. In fact your latex formatting here has the same mistake. Most underscores are missing! Instead of writing $\bigcup_{j=1}^m$ (`\biccup_{j=1}^m`), you wrote $\bigcup {j=1}^m$ (`\biccup {j=1}^m`).
>
>    It is not unclear, but a mistake. I require an explanation on this because the mistakes are so obvious that it should be immediate to anyone who looked at the equations.
>
> 2. Thanks for the clarification. However, the sentence you quote appears to be from the introduction, rather than Sec 3.2. I believe that Section 3.2 and/or 3.3 should be changed to make this clearer.
>
> 3. Thanks for the clarifications. I agree that the change will improve readability.
>
> 4. Thanks for the explanation. It appears that indeed, while there are similarities (i.e., fitting a simple model over summary stats), there are also important differences.
>
> 5. From just looking at the definitions, it is obvious that the definition of density (Eqn 3) is a scaled version of recall (Eqn 2). One of your definition is wrong.
>
> 6. To avoid confusion, I suggest not saying that you proposed these metrics, but you novelly adopted them for anomaly detection.
>
> 7. Thank you for the clarification. The proposed change sounds good to me.
>
> 9. Thank you for the pseudocode. It confirms my speculation and I think would be a strong addition to the paper.

---

> > ### Author Response · Authors · 2024-11-24
> >
> > Dear Reviewer GdRX,
> >
> > We are grateful to you again for your feedback, it has made our paper better.
> >
> > [In reference to points 1 & 5] After further review of your comments and our paper, we realized that there were, in fact, several formatting errors (overlooked due to a tooling glitch) with the LaTeX code which had caused it to not produce subscripts properly. We apologize for not noticing it after the first review, and we have fixed these errors. We thank you for catching this mistake. We have also decided that our past presentation of Figure 1 did not depict the density statistic optimally, and have modified it to focus on the reference points rather than the test points. We also noticed that the reference and test points were swapped in Equation 3 (which gave the impression that the density was a scaled recall) and have fixed this mistake as well. We also realize that we had misinterpreted your prior concern about $nearest_k$ to have been referring to $NND_k$, when you were referring to a $nearest_k$ we had used in the text of our description of the density statistic, and have replaced $nearest_k$ with $k$ to rectify this.
> >
> > - To fix concern [2] the statement has been added to Section 3.2.
> > - For [3] : The necessary changes have been made in Section 3.
> > - For [4] A dedicated paragraph has been added to Discussion (Section 6)
> > - For [6] The necessary change made to section 3.2
> > - For [7] Done. In Section 1, the first point inside the contribution has been fixed.
> > - For [8] Done. The pseudocode has been added to Appendix F.
> >
> > All changes made have been highlighted in blue text, for your ease of review.
> >
> > Moreover, to demonstrate our commitment to correctness and reproducibility, we attach an anonymized version of the codebase to run Forte. Altogether, we hope that these changes have made this section much clearer, and we thank you for helping us to make these improvements.
> >
> > To expand on the differences between Forte and DoSE, we do not use generative models in Forte, opting instead to use self-supervised representations. By not training additional models, we improve efficiency over DoSE, which required training an additional model in order to compute statistics for the detector. We further introduce a novel set of representation-based summary statistics, inspired by statistics used in manifold estimation (Naeem et al, 2020), which are useful in making a local measurement of the proximity of a query point to the data manifold. Thus, while we opted to build on top of DoSE (leveraging their insight into how to chain multiple statistics together to compose an OOD detector), the actual specifics of our model differs significantly. These differences have major impact: Forte significantly outperforms DoSE, which achieved 0.57 AUROC on CIFAR-10 vs CIFAR-100 (compared to 0.97 for Forte). In addition to outperforming DoSE, Forte also achieves SOTA performance, both when compared both against other post hoc methods (KNN gets 0.9 AUROC in Zhang et al. 2024) but also against methods which require additional training (RotPred gets 0.93 AUROC in Zhang et al. 2024), and against methods which are given known OOD points (e.g. Outlier Exposure gets 0.9 AUROC in Zhang et al. 2024). In all cases, this represents a more than 50% reduction in the total outstanding area under the ROC curve. (97% relative to DoSe, 70% relative to KNN/OE, and 57% relative to RotPred). We have added Figures 15 & 16 in the appendix to contextualize our performance compared to our peer methodologies.
> >
> > With these fixes, additions, and clarifications, we would like to ask if you have remaining concerns that have not yet been addressed. We are grateful for the opportunity to clear up any additional misunderstandings and improve the paper, and are excited to continue the discussion.
> >
> > We kindly request that you consider our explanations in your evaluation to increase our ratings and are open to any further questions or suggestions you may have.

---

> > > ### Author Response · Authors · 2024-11-27
> > > **Thank you note!**
> > >
> > > Dear Reviewer GdRX,
> > >
> > > Thank you for increasing our score from 3 to 6! We greatly appreciate your thorough review and constructive feedback throughout this process. Your attention to detail, particularly regarding the LaTeX formatting in mathematical definitions, has improved the clarity of our paper.
> > >
> > > Please let us know if there are any additional details or clarifications we can provide to further strengthen the paper.
> > >
> > > **For the meta-reviewers and area chairs**: We have carefully addressed all concerns raised during the review process, including mathematical formulation corrections, clearer comparisons with DoSE, and enhanced experimental validations through additional figures in the appendix.
> > >
> > > Under the thread by Reviewer 2YAU, reviewer GdRX has written
> > >
> > > ```
> > > The authors' response and revisions have significantly improved the paper. My concerns about the methods are mostly addressed.
> > > ```
> > >
> > >
> > >
> > >
> > > Best regards,
> > >
> > > The Authors

---

### Official Review · Reviewer_niAs · 2024-11-06

**Soundness:** 4
**Presentation:** 4
**Contribution:** 4
**Rating:** 10
**Confidence:** 4

**Summary:**

The authors present a methodology, Forte, which enables the identification of outliers using a rigorous unsupervised algorithm.  The algorithm relies on establishing metrics to identify which samples are in-distribution vs out-of-distribution.  The method is very general, and applicable to many models.  The authors use SVM, KDE, and GMM models with their method, and show that it is capable of distinguishing between real and synthetic data.

**Strengths:**

The paper is very well written, easy to follow, and highly implementable.  An extensive appendix provides supporting information and data.

**Weaknesses:**

None

**Questions:**

None.  The appendix cleared them up

---

> ### Author Response · Authors · 2024-11-13
> **Note of thanks**
>
> We are truly grateful for your detailed evaluation and positive feedback. We appreciate your recognition of our paper's clarity, implementability, and the comprehensive supporting materials we provided in the appendix. Your suggestion for conference highlighting is very much appreciated.

---

### Meta-Review · Area_Chair_Zz26 · 2024-12-21

**Metareview:**

The paper focuses on outlier detection, and is well received by all reviewers. There were concerns regarding the clarity and novelty, however, these were well addressed, even raising scores significantly during the rebuttal phase. Thus, I recommend acceptance.

**Additional Comments On Reviewer Discussion:**

There were no significant comments or changes during the reviewer discussion.

---

### Decision · Program_Chairs · 2025-01-22

Accept (Poster)